# Effects of a blend of essential oils in milk replacer on performance, rumen fermentation, blood parameters, and health scores of dairy heifers

Joana Palhares Campolina[1]☯, Sandra Gesteira Coelho[1]☯, Anna Luiza Belli[1]☯, Fernanda Samarini Machado[2‡], Luiz Gustavo R. Pereira[2‡], Thierry R. Tomich[2‡], Wanessa A. Carvalho[2‡], Rodrigo Otávio S. Silva[3‡], Alessandra L. Voorsluys[4‡], David V. Jacob[4‡], Mariana Magalhães Campos[2]☯*

**1** Department of Animal Science, Veterinary School, Federal University of Minas Gerais, Belo Horizonte, Minas Gerais, Brazil, **2** Brazilian Agricultural Research Corporation—EMBRAPA, National Center for Research on Dairy Cattle, Juiz de Fora, Minas Gerais, Brazil, **3** Department of Veterinary Science, Veterinary School, Federal University of Minas Gerais, Belo Horizonte, Minas Gerais, Brazil, **4** Adisseo, Campinas, São Paulo, Brazil

☯ These authors contributed equally to this work.
‡ These authors also contributed equally to this work.
* mariana.campos@embrapa.br

## Abstract

The aim of this study was to evaluate how the inclusion of a blend of essential oils in milk replacer (MR) affects different outcomes of dairy heifers. The outcomes evaluated: feed intake, performance, body development, blood cells and metabolites, insulin-like growth factor-1 (IGF-1), rumen fermentation, fecal scores, and respiratory scores. All outcomes were evaluated during pre-weaning (4–60 d of age), and carry-over effects during post-weaning (61–90 d of age) periods. The experimental units utilized were 29 newborn Holstein × Gyr crossbred dairy heifers, with genetic composition of 5/8 or more Holstein and 3/8 or less Gyr and body weight (BW) at birth of 32.2 ± 5.2 kg. Experimental units were assigned to either a control (CON, n = 15) or a blend of essential oil supplementation (BEO, n = 14) treatment, maintaining a balance of genetic composition. The BEO was supplemented in the MR with 1 g/d/calf of a blend of essential oils (Apex Calf, Adisseo, China) composed by plant extracts derived from anise, cinnamon, garlic, rosemary, and thyme. During the pre-weaning phase, all heifers were fed 5 L of MR/d reconstituted to 15% (dry matter basis), divided into two equal meals. Water and starter were provided *ad libitum*. During the post-weaning, animals received a maximum of 3 kg of starter/d, and *ad libitum* corn silage, divided into two meals. Feed intake, fecal and respiratory scores were evaluated daily. The BW was measured every three days, while body development was recorded weekly. Blood samples were collected on 0, 30, and 60 d of age for total blood cell count, weekly and on the weaning day to determinate ß-hydroxybutyrate, urea and glucose, and biweekly for IGF-1. Ruminal parameters (pH, volatile fatty acids, ammonia-N, and acetate:propionate proportion—C2:C3) were measured on days 14, 28, 42, 60, 74 and 90. A randomized complete block design with an interaction between treatment and week was the experimental method of choice to test the

**Data Availability Statement:** All relevant data are within the paper and its Supporting Information files.

**Funding:** This paper was financed by Brazilian Agricultural Research Corporation (EMBRAPA), Embrapa Dairy Cattle for funding this research and providing its facilities, and Adisseo Company for funding this research (project number: 20500.18/0005-2). Conselho Nacional de Desenvolvimento Científico e Tecnológico (CNPq, Brasília, Brazil), Instituto Nacional de Ciência e Tecnologia Ciência Animal (INCT, Viçosa, Brazil) for granting the scholarship. Alessandra L. Voorsluys and David V. Jacob received salary from Adisseo Company. The specific roles of these authors are articulated in the 'author contributions' section. The funders did not play any role in the study design, data collection and analysis, decision to publish, or preparation of the manuscript and only provided financial support.

**Competing interests:** This paper was financed by Brazilian Agricultural Research Corporation (EMBRAPA), Embrapa Dairy Cattle for funding this research and providing its facilities, and Adisseo Company for funding this research (project number: 20500.18/0005-2). Conselho Nacional de Desenvolvimento Científico e Tecnológico (CNPq, Brasília, Brazil), Instituto Nacional de Ciência e Tecnologia Ciência Animal (INCT, Viçosa, Brazil) for granting the scholarship. Alessandra L. Voorsluys and David V. Jacob received salary from Adisseo Company. There are no patents, products in development or marketed products to declare. This does not alter our adherence to PLOS ONE policies on sharing data and materials.

hypothesis of the BEO's effect on all outcomes. An ANOVA procedure was used for continuous outcomes, and a non-parametric test was used for the ordered categorical outcomes, both adopting a CI = 95%. Results indicated that there was not enough evidence to accept the alternative hypothesis of the effect of BEO in MR on feed intake, performance, body development, and blood metabolites during both pre-weaning and post-weaning periods. However, results indicated that the inclusion of BEO in MR significantly affects the proportion of C2:C3 during pre- and post-weaning ($P = 0.05$). Similarly, the effect was significant for basophil ($P \leq 0.001$), and platelet ($P = 0.04$) counts pre-weaning. The interaction between week and treatment was also significant for lymphocytes ($P \leq 0.001$), revealing a cumulative effect. Lastly, fecal scores were also significant ($P = 0.04$) during pre-weaning, with lower values for BEO. The BEO contributed to ruminal manipulation in pre-weaning and carry-over effects in post-weaning, immunity improvement, and decreased morbidity of neonatal diarrhea in the pre-weaning phase.

## Introduction

A good calf-rearing program should embrace aspects that encompass from body development, stress reduction, meet nutritional requirements, and housing management to optimize calf health status. Average daily gain (ADG) and body weight (BW) at weaning are key metrics used to measure the success of the rearing program. It is well known that these parameters are related to the success of the rearing program, as well as the heifer's future milk production. Therefore, a bad life start can negatively impact animal adult performance [1]. Nutritional problems and neonatal diseases, especially diarrhea and respiratory syndrome, are some examples of negative impacts on the calf's young life. They can act as stressors, lowering calf immunity, increasing animal susceptibility to other disorders, and raise mortality rates [2, 3].

Therefore, tools that help provide proper nutrition, and improve heifer development and health, are essential to reduce disease morbidity and mortality and accelerate the calf development. Additionally, since a calf is born functionally as a non-ruminant, the digestive system, and other organs and tissues, change in several weeks and the microbiota colonization changes to adapt to these transformations [1]. The bacteria in the rumen must start the fermentation of carbohydrates, so the calf can become dependent mostly on volatile fatty acids (VFA) and not on lactose-driven metabolism [4]. For that matter, procedures that reduce the animal's susceptibility to pathogens and stressors, and help this pathway change, may improve future performance and productivity [5].

Since the discovery of the improvement in animal growth due to antibiotics almost 80 years ago, antibiotic growth promoters (AGP) have been widely used as a tool to improve both rumen development and animal health [5, 6], prevent diseases, and increase performance and feed efficiency [7, 8]. However, the use of AGP in animal production for these purposes has been under severe criticism and banned in several countries [9]. The overuse of antimicrobial's concerns human health since there is already a well-established correlation between the increase of bacterial population resistance and the use of AGP, putting both humans and animals at risk [10]. The World Health Organization considers the antimicrobial resistance one of the three major threats to public health [11]. However, the global trends in antimicrobial use show that some countries with the largest share of global antimicrobial consumption in food animals initiated a shift toward a more conservative use [12]. The EU banned the use of

AGP since 2006 [13] and the US published the Veterinary Feed Directive in 2015, which limited the use of AGP under the professional supervision of a licensed veterinarian [14] and banned all medically important antimicrobials for humans in 2017 [11]. Other big livestock producing countries, such as China and Mexico, are also changing the acceptability of AGP's use in food animal production [11]. Therefore, there is a motivation for more prudent use of antimicrobials [15] and research for substitutes that can improve animal performance and health. A large number of new additives such as prebiotics and probiotics, organic acids, phytogenic substances, and essential oils have shown good results to improve animal production [4, 16] and appear to be a good alternative to decrease the use of AGP and alleviate the antimicrobial resistance [16, 17]. One of these alternatives is the phytogenic feed additives, also known as phytobiotics and botanicals, commonly defined as plant secondary compounds [18, 19].

Essential oils are one of the additives derived from herbal plant secondary chemical components. They are constituted by volatile or ethereal oils that have been applied as a natural and safe alternative for antibiotics [20]. Some of their properties are antiseptic and antimicrobial activities that interfere with bacterial, fungal, and protozoa cell functioning [16], presenting a similar efficiency to treat some diseases as antibiotics [21]. They also contribute to the prevention of oxidative stress [22] and help the immune response change leukocyte phagocytic activity and inhibit the complement system [23]. Lastly, essential oils have been shown to function similarly to ionophores, a type of AGP [24]. They can influence gastrointestinal tract development, rumen microbiological activity, improve feed efficiency, and decrease neonatal diseases [16, 25].

Studies focusing on essential oils' action as growth promotors for pigs and poultry show the supplementation's positive effects, generally associated with effects on the gastrointestinal tract (GIT) [26, 27]. In those species, essential oil supplementation increased digestibility, improved pancreatic enzymes' activity, changed microbiota, impacted the absorption of amino acids in the intestines, and, consequently, feed conversion rate [27–29]. The supplementation also increases immunoglobulins levels and immune response [30], decreases specify pathogens concentrations in feces [31, 32] and presented an insecticidal [33], acaricidal and antioxidant effects [34]. However, there is inconsistent data between other species, probably explained by the complexity of the essential oils' molecules and differences among the many types of GIT [19]. Previous studies have shown that essential oils supplementation in calf's solid starter improves performance [35, 36], rumen fermentation [37], and diarrhea severity [38]. However, the effects on liquid diet supplementation are scarce.

This study aimed to evaluate if the supplementation of a commercial blend of essential oils (BEO) in milk replacer (MR) affects feed intake, performance, feed efficiency, body development, blood cells and metabolites, insulin-like growth factor-1 (IGF-1), ruminal parameters, fecal and respiratory scores of dairy heifers during pre-weaning and post-weaning periods. We hypothesized that BEO supplementation in MR during pre-weaning would improve performance and positively influence blood parameters and health scores of dairy heifers.

## Material and methods

Protocols for this study were approved by the Ethics Committee of Embrapa Dairy Cattle (protocol number 9078250118). The experiment was conducted on the Embrapa Dairy Cattle Experimental Farm, located in Coronel Pacheco, Minas Gerais, Brazil, from March to September 2018.

### Animals, treatments, and management

Twenty-nine newborn Holstein × Gyr crossbred dairy heifers, with genetic composition of 5/8 or more Holstein and 3/8 or less Gyr and BW at birth of 32.2 ± 5.2 kg, were used and equally

distributed among treatments. They were separated from their dams immediately after birth and moved to individual sand-bedded pens (1.25 × 1.75 m, tethered with 1.2 m long chains), allocated in a barn with open sides and end-walls.

All heifers received 10% of their BW of good quality colostrum (Brix > 23%) before 6 h after birth and had their umbilical cord immersed in an iodine solution (10%).

From 2 to 3 d of age, heifers were fed 5 L/d of transition milk divided into two equal meals offered at 0800 and 1600 h, in buckets provided with rubber teats (Milkbar, New Zealand). At 3 d of age, blood samples were collected via jugular venipuncture with a clot activator tube (Labor Import, Osasco, Brazil). They were left at room temperature for 30 min and then centrifuged at $1,800 \times g$ for 10 min (22–25°C). The serum was piped into a Brix refractometer (Aichose refractometer, Xindacheng, Shandong, China) to measure the success of the passive immune transfer. Heifers were enrolled only if the Brix was higher than 8.4%.

Water and commercial calf starter (Soymax Rumen pre-inicial Flocculated, Total Alimentos, Três Corações, Brazil, Table 1) were offered in buckets for ad libitum intake (10% orts of solid feed).

At 4 d of age, heifers were assigned to one of two experimental treatments maintaining a balance of the birth month, birth BW, genetic composition, and % Brix value. They were fed at 5 L/d of an MR (Kalvolak, Nutrifeed, Netherlands; Table 1) reconstituted at 15% (dry matter basis), divided into two equal meals (0800 and 1600 h) into buckets provided with rubber teats (Milkbar). The experimental treatments were: Control, no additive (CON; n = 15), and a commercial blend of essential oils additive supplemented at a rate of 1 g/d/calf (BEO, Apex Calf, Adisseo, China; n = 14), as recommended by the manufacturing company. The blend of essential oils is a dry powder that contains a mix of plant extracts derived from anise, cinnamon, garlic, rosemary, and thyme. The amount of the additive for each meal was weighed to have 0.5 g and kept in 15 mL tubes in a dark box. They were then mixed with a 10 mL of MR, homogenized, and incorporated in 0.49 L of MR (0.5 g/calf at morning meal and 0.5 g/calf at afternoon meal) to ensure total ingestion of the product. Immediately after ingesting 0.5 L MR with 0.5 g of the blend of essential oils, the rest of the meal was given. One person was responsible for refilling the milk bucket as soon as the animals had finished, so it would not change the ingestion rate. This person would also evaluate MR acceptance.

Table 1. Nutrient composition (% DM basis ± SD) of Milk Replacer (MR), starter, and corn silage.

| Item | MR[1] | Starter[2] | Corn Silage |
|---|---|---|---|
| DM (%) | 96.0 ± 0.4 | 86.7 ± 0.7 | 36.1 ± 3.1 |
| CP (% of DM) | 19.4 ± 0.5 | 17.1 ± 0.5 | 7.9 ± 0.7 |
| Ether extract (% of DM) | 14.1 ± 0.6 | 3.9 ± 1.2 | 4.3 ± 0.5 |
| Organic Matter (% of DM) | 9.7 ± 0.2 | 7.2 ± 1.5 | 6.0 ± 1.1 |
| NDF (% of DM) | – | 22.1 ± 2.9 | 46.1 ± 4.1 |
| ADF (% of DM) | – | 10.6 ± 0.9 | 28.9 ± 3.5 |
| Gross Energy (Mcal/kg of DM) | 4.5 ± 0.1 | 4.3 ± 0.1 | 4.5 ± 0.1 |

[1] Powder integral milk, wheat isolated protein, acidifying additive, whey, coconut oil, palm oil, vitamin A, Vitamin D3, Vitamin E, Vitamin C (Kalvolak, Nutrifeed, Netherlands).

[2] Basic composition: oats (rolled grains), calcitic limestone, sodium chloride, corn gluten meal, defatted corn germ, wheat bran, soybean meal, rice hulls, kaolin, molasses, flocculated corn, ground corn, corn grain, alfalfa hay, monensin, citrus pulp, dried sugarcane yeast, whole toasted soybean, sodium selenite, copper sulfate, manganese sulfate, cobalt sulfate, iron sulfate, zinc sulfate, calcium iodate, vitamin A, vitamin B1, vitamin B12, vitamin B2, vitamin B6, vitamin C, vitamin D3, vitamin E, vitamin K, niacin, pantothenic acid, folic acid, biotin, propionic acid, caramel aroma, milk aroma, and probiotic additive.

Heifers were weaned abruptly at 60 d of age. During the post-weaning period, from 61 to 90 d of age, all heifers received starter and corn silage (Table 1). The amount of corn silage provided was enough to result in at least 10% orts, and the starter intake was fixed for a maximum of 3.0 kg calf/d, divided into two meals. All heifers were dehorned at 70 d of age and received local anesthesia (5.0 mL/horn, Lidovet, Bravet, Engenho Novo, Brazil) and 2 d of non-steroid anti-inflammatory treatment (0.025 mL/kg, Maxicam 2%, Ouro fino, Cravinhos, Brazil).

## Intake and nutritional composition analysis

Feed intake (MR, starter, water, and corn silage) were measured daily. Samples of MR, starter, and corn silage were collected three times a week to obtain a weekly pool for nutritional analyses. Samples of starter and corn silage were oven-dried at 55˚C for 72 h and ground in Wiley mill (model 3, Arthur H. Thomas Co., Philadelphia, PA) through a 1-mm screen before analysis. Starter, corn silage, and MR were analyzed to determine DM (Method 934.01), CP (Method 988.05), ether extract (Method 920.39), ash (Method 942.05), according to AOAC [39]. The concentrations of NDF and ADF were determined in sequence using the method described by Van Soest et al. [40]. Gross energy was determined using an adiabatic bomb calorimeter (Parr Instrument Company, Moline, IL).

## Structural growth

Body weight (BW) was measured on the day of birth, 3 d of age, and, after that, every 3 d before the morning meal using a weighing-machine (ICS 300, Coimma, Dracena, Brazil). Wither height (distance from the base of the front feet to the withers), rump height (distance from the base of the rear feet to the rump), rump width (distance between ileus), and heart girth (circumference of the chest) were measured on the day of birth and, after that, every 7 d until the end of the experiment. These measurements were taken on a flat surface using a portable hypometer and a measuring tape. Feed efficiency was calculated using the ADG and DMI ratio [41].

## Rumen fermentation

Rumen fluid samples were collected through an oroesophageal tube 4 h after morning feeding at 14, 28, 42, 60, 74, and 90 d of age, and pH was assessed using a portable potentiometer (Phmetro T-1000, Tekna, Araucária, Brazil). Two aliquots of 10 mL of ruminal fluid were separated. One was acidified with 1 mL of 20% metaphosphoric acid, and the other with 2 mL of 50% sulfuric acid. These samples were stored at -20˚C for further analysis of VFA and nitrogen ammonia. Nitrogen ammonia concentration was quantified using the colorimetric distillation method proposed by Chaney and Marbach [42]. Its absorbance was measured at 630 nm (Thermo Fisher Scientific, Madison, WI, USA) after Kjeldahl distillation with magnesium oxide and calcium chloride according to Method 920.03 [39]. The VFA concentrations were determined in the samples previously centrifuged at $1,800 \times g$ for 10 min at room temperature (22–25˚ C) by high-performance liquid chromatography (Waters Alliance e2695 Chromatograph, Waters Technologies do Brazil LTDA, Barueri, SP, Brazil).

## Blood cell count, metabolites and IGF-1

Jugular blood samples were collected at birth before colostrum ingestion and, 3 h after morning feeding on days 0, 7, 14, 21, 28, 35, 42, 49, 56, 60, 67, 74, 81 and 90, for beta-hydroxybutyric acid (BHB), urea and glucose and, on days 0, 14, 28, 42, 60, 74 and 90, for IGF-1 concentrations. Blood samples were collected into tubes without anticoagulant (for BHB and urea), with

sodium fluoride (for glucose), or with heparin for IGF-1 (Labor Import, Osasco, Brazil). They were immediately transported on ice to the laboratory and were centrifuged at 3000 x *g* for 10 min at room temperature (22–25°C). Two aliquots of each metabolite and hormone sample were individually allocated into microtubes and frozen at -20°C for further analysis. The serum concentration of BHB and urea were determined by an auto-analyzer (Cobas Mira Plus, Roche Diagnostic Systems, Risch-Rotkreuz, Switzerland) using commercial kits (Ranbut-D-3-Hidroxibutyrate, Randox Laboratories Ltd., Antrim, UK; Urea UV, Kovalent do Brasil Ltda., Bom Retiro São Gonçalo, Brazil). Plasma glucose was measured in a microplate Spectrophotometer EON (Biotek Instruments Inc., Winooski, VT) using the enzymatic colorimetric method (Kovalent do Brasil Ltda., Rio de Janeiro, Brazil). The plasma concentrations of IGF-1 were analyzed using chemiluminescence assay (Immulite2000 Systems 1038144, IGF-1 200, Siemens Healthcare Diagnostics Products Ltd., Llanberis, Gwynedd, UK).

Blood samples were collected for complete blood count during preweaning at 0, 30 and 60 d of age, by jugular vein puncture into EDTA tubes (Labor Import, Osasco, Brazil), and immediately transported on ice to the laboratory. An automatic hematology cell counter (SDH– 3 vet, Labtest Diagnóstica S.A., Brazil) was used to evaluate: red blood cell count (RBC), packed cell volume (PCV), hemoglobin (Hb), mean corpuscular volume (MCV), mean corpuscular hemoglobin concentration (MCHC), platelet and total white blood cell count. Manual white cell blood differential counting was also performed by microscopic examination evaluating 100 leukocytes in a 1,000 x microscopic magnification for total leukocyte count, basophils, eosinophils, neutrophils, band neutrophils, segmented neutrophils, lymphocytes, monocytes. Morphological changes, such as toxic neutrophils, reactive lymphocytes, and activated monocytes, were calculated [43]. In addition, platelet to lymphocytes ratio (PLR) and neutrophils to lymphocytes ratio (NLR) were calculated.

## Health measurements

Health measurements (fecal and respiratory scores) were performed daily, in the morning, before other animal management. Fecal scores were graded according to the University of Wisconsin calf health scoring chart [2], as follows: 0 –normal (firm but not hard); 1 –soft (does not hold form, piles but spreads slightly); 2 –runny (spreads readily to about 6 mm depth); and 3 –watery (liquid consistency, splatters). A heifer was considered to have diarrhea if the fecal score was 2 or 3. Severe diarrhea was considered when the fecal score was 3.

Daily respiratory score evaluations were adapted from the University of Wisconsin calf health scoring chart [2], considering rectal temperature score: 0 –temperature between 37.8 and 38.3°C, 1 –temperature between 38.4 and 38.8°C, 2 –temperature between 38,9 and 39.3°C, 3 –temperature above 39.4°C; cough score: 0 –none, 1 –induce single cough, 2 –induced repeated or occasional spontaneous coughs, 3 –repeated spontaneous coughs; nose score: 0 –normal serous discharge, 1 –small amount of unilateral cloudy discharge, 2 –bilateral cloudy or excessive mucus discharge, 3 –copious bilateral mucopurulent discharge; eye score: 0 –normal, no discharge, 1 –small amount of ocular discharge, 2 –moderate amount of bilateral discharge, 3 –heavy ocular discharge; ear score: 0 –normal, 1 –ear flick or head shake, 2 –slight unilateral drop, 3 –head tilt or bilateral drop. A final respiratory score was determined by the summation of temperature, cough, nose, eye, and ear scores.

Heifers were treated with non-steroid anti-inflammatory (0.025 mL/kg, Maxicam 2%, Ouro fino, Cravinhos, Brazil) when respiratory score sum was above 4, or if they presented fever for two consecutive days. Fever was considered when the pre-meal morning temperature was $\geq$ 39.4°C. One dose of enrofloxacin antibiotic (0.075 mL/kg, Kinetomax, Bayer, São Paulo, Brazil) was administered when a pulmonary commitment was detected (shortness of

breath, edema and/or crepitation detected by auscultation) or an animal had fever combined with diarrhea for 2 d subsequently.

## Minimum inhibitory concentration

The broth dilution method was used to evaluate the minimum inhibitory concentration (MIC) of BEO against two relevant enteric bacteria: enterotoxigenic *Escherichia coli* (K99[+] strain) and *Salmonella typhimurium* previously isolated from an outbreak in calves [44]. Two different preparations of BEO product were used to perform MIC: a—homogenized in purified water; b —homogenized in a solution with 3.0 g of isopropyl myristate, 8.25 g of propylene glycol, 7.25 g of Tween 80 (Sigma-Aldrich, Santo André, Brazil) and 100 mL of water. Both preparations were submitted to 0.22 µm filtration. A solution with an initial concentration of 1.0 mg/mL was submitted to serial dilutions from 1:2 to 1:256 in 96-wells plates. Thus, 100 µL of a solution containing $5 \times 10^5$ CFU/mL of the two selected bacteria. After overnight incubation at 35˚C, microtiter plates were examined for visible bacterial growth evidenced by turbidity and color change.

## Statistical analysis

Statistical analysis was conducted utilizing R[®] (R Core Team, 2019). The data collected was summarized by period (pre-weaning– 4 to 60 d and post-weaning– 61 to 90 d) and per week within each period. A randomized complete block experimental design with repeated measures was implemented to test the hypothesis of the effect of the blend of essential oils on each performance outcome. More specifically, the outcomes analyzed were feed intake, structural growth, ruminal, blood, and health parameters. The control treatment was assigned 15 experimental units (CON), while the blend of essential oils supplementation treatment was assigned 14 (BEO).

The analysis of each outcome was performed independently of all others using linear mixed models (package: nlme). Each independent outcome was modeled as a function of the following fixed effects: treatment, experimental week, the interaction between treatment and week. The genetic composition of the animal was included as a blocking effect. Birth month, birth body weight and Brix value were assessed only to verify if the animals were homogeneously distributed but were not used as a blocking effect. Birth weight and serum Brix value were tested as a covariate but did not improve statistical significance. Therefore, they were eliminated from the model. The effect of heifer within treatment was included in the models to account for individual variability.

The continuous outcomes such as intakes, structural growth, ruminal, and blood parameters were analyzed with ANOVA. A 95% Confidence Interval was adopted to verify the null hypothesis, and *P*-values were produced with a Fisher test. All outcomes were tested for normality to meet the required assumptions of this model, and a variable transformation was applied to milk replacer intakes to meet that assumption.

The categorical outcomes fecal and respiratory scores were analyzed using a non-parametric aligned rank transformation test, implemented in the R package ARTool. A 95% Confidence Interval was also adopted for the non-parametric tests. Associations between the fecal scores and MR intakes were assessed by using the Spearman correlations.

## Results and discussion

### Intake and heifer performance

Most studies evaluate essential oils or a supplement with BEO to dairy calves, feed the additive in the starter to benefit rumen development, and accelerate growth. However, the intake of

starter in the first weeks of age is small [45], and the timing of the occurrence of enteric diseases is mainly on the first 30 days of life [2]. Due to the calf's limited capability of ingesting large solid feed amounts in the first days of life, the supplement intake in the starter could be limited, and the desired supplementation level may not be achieved based on intake levels of the starter. Therefore, in this trial, BEO was offered in the liquid diet since the aim was to verify if it would impact on disease morbidity and gut development, and subsequently, on animal's performance.

The supplemented heifers consumed the same amount of liquid diet as the control group, indicating no ingestibility issues of BEO (Table 2). Differences described in the literature between flavor and palatability of BEOs could be due to the delivery method, as well as essential oil plant sources and extraction process [16]. Studies using different supplemented types of essential oils to other animal species' reported different preferences and acceptability of these essential oils, with changes among animal species and category, juvenile x adults [19]. Previous work with weaned heifers supplemented with cinnamaldehyde essential oil in a total mix ration showed a preference in the taste of the ration without additive. This supplementation caused a change of feed intake, and it was related to palatability problems with the essential oil used in the experiment [46]. However, although cinnamon is an ingredient that is in the mixture in our study, we did not run a palatability test to verify this outcome. It must be point also that the additive was given mixed with a small amount of MR to allow complete ingestion. Visually, the time on ingestion was the same, and all the calves consumed all MR. Therefore, ingestibility of the mixture was not a problem, However, further tests with essential oils palatability to dairy calves are needed.

Although there were no differences between MR intake between treatment and the given amount was fixed, there were a week effect and a week and treatment interaction effect ($P \leq$ 0.001, Table 2, Fig 1). From the end of week 1 until week 3, heifers had diarrhea and this event impacted on MR intake, since intake decrease when animals are sick. Differences between

**Table 2. Pre and post-weaning Milk Replacer (MR) intake, starter intake, total dry matter intake (DM), total crude protein intake (CP), total gross energy and water intake of heifers of control (CON) and supplemented with blend essential oils (BEO) in milk replacer during pre-weaning.**

| Intake | Treatment | | SEM | P–value[3] | | |
|---|---|---|---|---|---|---|
| | CON[1] (n = 15) | BEO[2] (n = 14) | | T | W | T x W |
| **Pre-weaning (4 to 60 d)** | | | | | | |
| MR (kg of DM/d)[4] | 0.71 (0.705–0.721) | 0.71 (0.701–0.716) | - | 0.30 | <0.001 | <0.001 |
| Starter (kg of DM/d) | 0.30 | 0.31 | 0.02 | 0.92 | <0.001 | 0.82 |
| Total DM (kg/d) | 1.00 | 1.16 | 0.06 | 0.58 | <0.001 | 0.31 |
| Total CP (kg/d) | 0.19 | 0.19 | 0.01 | 0.58 | <0.001 | 0.31 |
| Total gross energy (Mcal/kg) | 4.51 | 4.59 | 0.12 | 0.58 | <0.001 | 0.30 |
| Water (kg/d) | 1.39 | 1.30 | 0.32 | 0.98 | <0.001 | 0.64 |
| **Post-weaning (61 to 90 d)** | | | | | | |
| Starter (kg of DM/d) | 1.84 | 2.02 | 0.28 | 0.39 | <0.001 | 0.31 |
| Corn Silage (kg of DM/d) | 0.12 | 0.11 | 0.03 | 0.51 | <0.001 | 0.26 |
| Total DM (kg/d) | 1.97 | 2.14 | 0.29 | 0.39 | <0.001 | 0.32 |
| Total CP (kg/d) | 0.44 | 0.47 | 0.07 | 0.36 | <0.001 | 0.72 |
| Total gross energy (Mcal/kg) | 8.61 | 9.35 | 1.34 | 0.39 | <0.001 | 0.29 |
| Water (kg/d) | 5.41 | 5.69 | 0.84 | 0.61 | <0.001 | 0.10 |

[1]CON = control

[2]BEO = 1 g/calf/d blend of essential oil.

[3]T = treatment effect; W = week effect, T x W = treatment by week interactions.

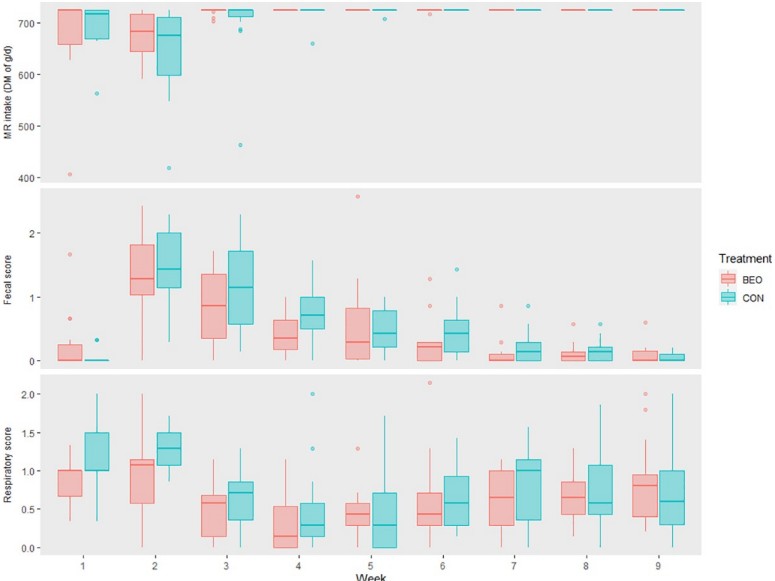

**Fig 1. MR intake (g of DM/d), respiratory and fecal scores of control heifers (CON) and heifers supplemented with 1.0 g/calf/ d of the blend of essential oils (BEO) in milk replacer during the pre-weaning period.**

treatments were observed in those weeks, with lower intake for the CON. An observed effect between fecal scores and MR intake was found ($P \leq 0.001$), besides a low correlation value (-0.25). Thus, results revealed a negative association between both parameters, where higher fecal scores reduced MR intake, and vice versa.

Intake of starter, water, total DM, CP, and gross energy, ADG and feed efficiency were not affected by treatment during pre- and post-weaning (Tables 2 and 3). A previous study tested a commercial blend of essential oils for dairy calves using two supplementation routes (MR and starter), and had similar results for intake, BW and ADG during preweaning [47]. However, other studies that also used a commercial source of essential oils in the starter found better ADG and feed efficiency during the preweaning period for supplemented calves, as well as higher BW during weaning [36, 37]. As for the carry-over effect on post-weaning in those studies, it has been observed that calves supplemented with essential oils in the starter had higher ADG and lower feed efficiency [48]. In our study, we did not find any carry-over effect on post-weaning for the performance outcomes.

In our study, the lack of differences in evaluated outcomes could be because of the supply route, dosage, or the essential oil plant sources and extraction process. It also must be highlighted that the starter provided contained monensin and other probiotic additives. They are important and efficient additives used not only as a growth promoter but also as coccidiosis control and prevention [49]. However, some studies believe that the combined supplementation of monensin and essential oils could mask the effect of the essential oils or even compete for the same mechanisms of action [50]. In this study, no antagonism between additives was observed, as there were no negative responses for BEO compared to CON. It must also be highlighted that monensin was provided in the starter and the essential oil in the milk replacer. Thus, they would act in different compartments, the rumen and the intestines. To better understand this interaction and a possible effect, it is necessary for other studies to evaluate the impact of the essential oil's supplementation with or without monensin, as also the mechanism of action of the different essential oils.

## Structural growth

Structural body growth was not affected by BEO supplementation in MR (Table 3) during pre- and post-weaning. As was also observed for intake and ADG, a week effect ($P \leq 0.001$) was detected in all variables due to healthy animal growth. It was previously suggested that essential oils supplementation could only be effective in structural growth when associated with higher protein concentration in the starter due to an interaction between protein level supplementation and essential oils supplementation [37]. Other studies suggested that feeding essential oils could enhance growth performance if fed at an appropriate rate and in a determined amount [36]. In our study, the calves were fed with protein levels to meet their requirements for optimal growth. However, we did not test different protein levels to see if this interaction could change structural growth. On the other hand, in other species, the increase in structural growth, as well as daily weight gain and feed conversion for supplemented animals, are generally related to a more mature and developed gut. This more developed gut helps the supplement to be absorbed more quickly, improving gut immunity and microbiota, and as a consequence, the animals' body growth [51].

## Rumen fermentation

There were no differences in ruminal pH for CON and BEO treatments during the pre-weaning period. Previous studies also did not find changes in ruminal pH for animals supplemented with essential oils [16, 26, 28]. During the post-weaning period, the BEO treatment presented

**Table 3. Pre- and post-weaning performance and structural growth of heifers of control (CON) and supplemented with essential oils blend (BEO) in milk replacer during pre-weaning.**

| Item | Treatment | | SEM | P–value[3] | | |
|---|---|---|---|---|---|---|
| | CON[1] (n = 15) | BEO[2] (n = 14) | | T | W | T x W |
| **Performance** | | | | | | |
| Birth BW (kg) | 32.40 | 31.97 | 0.59 | 0.85 | – | – |
| Weaning BW (kg) | 64.36 | 66.66 | 1.07 | 0.45 | – | – |
| Final BW (kg) | 89.88 | 93.34 | 1.57 | 0.57 | – | – |
| ADG preweaning (kg/d) | 0.55 | 0.53 | 0.02 | 0.49 | <0.001 | 0.23 |
| ADG postweaning (kg/d) | 0.81 | 0.84 | 0.27 | 0.76 | 0.001 | 0.60 |
| Feed efficiency preweaning (kg/kg) | 0.62 | 0.56 | 0.008 | 0.06 | <0.0001 | 0.29 |
| Feed efficiency postweaning (kg/kg) | 0.44 | 0.42 | 0.04 | 0.50 | 0.68 | 0.42 |
| **Body measures** | | | | | | |
| **Preweaning (4 to 60 d)** | | | | | | |
| Withers height (cm) | 72.74 | 72.59 | 1.25 | 0.86 | <0.001 | 0.48 |
| Rump height (cm) | 75.89 | 75.90 | 0.66 | 0.98 | <0.001 | 0.62 |
| Rump width (cm) | 19.03 | 19.42 | 0.66 | 0.23 | <0.001 | 0.94 |
| Heart girth (cm) | 80.70 | 81.50 | 0.009 | 0.34 | <0.001 | 0.68 |
| **Postweaning (61 to 90 d)** | | | | | | |
| Withers height (cm) | 82.66 | 82.55 | 1.06 | 0.92 | <0.001 | 0.72 |
| Rump height (cm) | 86.02 | 86.64 | 1.07 | 0.61 | <0.001 | 0.80 |
| Rump width (cm) | 22.59 | 22.99 | 0.43 | 0.28 | <0.001 | 0.40 |
| Heart girth (cm) | 96.55 | 97.85 | 1.44 | 0.27 | <0.001 | 0.40 |

[1]CON = control

[2]BEO = 1 g/calf/d blend of essential oil.

[3]T = treatment effect; W = week effect, T x W = treatment by week interactions.

a lower pH ($P$ = 0.05, Table 4). Since there were no differences between treatments during pre-weaning, the carry-over effect may not be assumed to be the answer to this difference. Although no differences in intake were observed, heifers' ingestion behavior might justify the difference in post-weaning pH. In other words, the amount of starter consumed before sampling and its impact on ruminal pH. However, this behavior was not evaluated since intake was measure only once every 24 hours.

Considering that low pH could enhance essential oils effects, this could benefit younger calves that are supplemented with essential oils in the starter [24]. It is also known that its supplementation is related to antimicrobial and antifungal effects [16, 24]. Essential oils cause hydrophobicity and disrupt bacteria membrane, increasing water permeability and causing a toxic effect on the microorganism [7, 12]. This activity could result in inhibition of ruminal deamination and methanogenesis [25]. This effect on the modulation of nitrogen path would result in a decrease of the ruminal nitrogen ammonia, methane and acetate concentrations and an increase of the propionate and butyrate concentrations [24].

Changes in these profiles in rumen fluid would also alter the acetate:propionate (C2:C3) proportions. Since butyrate and propionate are important for ruminal papillae development, and especially propionate is used in the gluconeogenesis route [5], a smaller C2:C3 ratio is wanted. In this experiment, BEO supplementation did not alter VFA values, but did reduced the C2:C3 proportion during the pre- ($P$ = 0.05) and post-weaning phases ($P$ = 0.006) (Table 4). Confirming these findings, previous studies registered a lower C2:C3 proportion for calves in both groups supplemented with essential oils in the starter (1.56 and 1.47) compared with two control groups (2.02 and 1.77) [37]. On the other hand, reports are not always constant in the literature, since higher C2:C3 proportion for pre-weaning calves supplemented with thyme essential oils (2.25 x 1.78) were already reported [52]. Despite our findings, it must

**Table 4. Pre- and post-weaning rumen mean values of rumen pH, ammonia nitrogen (Ammonia-N) and volatile fatty acids (VFA) of control heifers (CON) and heifers supplemented with essential oils blend (BEO) in milk replacer during pre-weaning.**

| Item | Treatment | | SEM | P–value[3] | | |
|---|---|---|---|---|---|---|
| | CON[1] (n = 15) | BEO[2] (n = 14) | | T | W | T x W |
| **Pre-weaning (4 to 60 d)** | | | | | | |
| Rumen pH | 5.99 | 5.85 | 0.52 | 0.37 | 0.03 | 0.06 |
| Rumen ammonia-N (mg/dL) | 11.40 | 13.80 | 0.03 | 0.15 | <0.001 | 0.37 |
| Rumen VFA (µmol/mL) | | | | | | |
| Acetic (C2) | 30.80 | 27.16 | 8.15 | 0.24 | <0.001 | 0.14 |
| Propionic (C3) | 18.88 | 20.01 | 7.11 | 0.59 | <0.001 | 0.14 |
| Butyric (C4) | 0.80 | 0.80 | 0.08 | 0.83 | 0.005 | 0.98 |
| C2:C3 | 1.97 | 1.69 | 0.12 | 0.05 | <0.001 | 0.95 |
| **Post-weaning (61 to 90 d)** | | | | | | |
| Rumen pH | 6.19 | 5.90 | 0.001 | 0.05 | 0.001 | 0.86 |
| Rumen ammonia-N (mg/dL) | 10.97 | 9.53 | 9.03 | 0.17 | 0.91 | 0.88 |
| Rumen VFA (µmol/mL) | | | | | | |
| Acetic (C2) | 38.32 | 39.03 | 8.48 | 0.81 | 0.006 | 0.93 |
| Propionic (C3) | 28.27 | 30.69 | 5.16 | 0.41 | 0.003 | 0.75 |
| Butyric (C4) | 5.94 | 6.16 | 1.19 | 0.82 | 0.95 | 0.62 |
| C2:C3 | 1.43 | 1.23 | 0.20 | 0.006 | 0.74 | 0.93 |

[1]CON = control

[2]BEO = 1 g/calf/d blend of essential oil.

[3]T = treatment effect; W = week effect, T x W = treatment by week interactions.

be highlighted that, in our experiment, essential oils were provided mixed in small amounts of MR to ensure the whole intake of the product. If the BEO was provided in the starter, changes in the rumen would be expected. By providing the BEO in the MR, the treatment should bypass the rumen and have minimal impact on local ruminal microbiota and VFA. Nevertheless, since the MR amount was small and given at the beginning of the feeding, one hypothesis could be that the esophageal groove was still open, permitting essential oils content to arrive at the rumen. Another hypothesis could be a potential communication from the intestines and the forestomach were the nutrients on the lower gut caused adaptations on the upper gut, improving its function and growth, as well as nutrient use and differences in VFA proportions [53]. In monogastric animals, supplementation of essential oils has shown a direct effect on the gut microflora and effects on the gut-associated immune system, causing positive changes in nutrient digestibility and animal performance [54]. A third theory to explain the changes in C2:C3 is that the changes in rumen could not be only by the BEO supplementation, but the interaction between the BEO and the monensin in the starter. They have a similar mechanism of actions and could cause the increase in propionate in the rumen, not enough to be seen when evaluating the VFA alone, but shifting ruminal fermentation and cause differences in C2:C3 proportions [50].

However, despite changes in C2:C3 proportions, nitrogen ammonia concentrations were not affected by BEO supplementation during pre- and post-weaning (Table 4). Previous studies reported higher nitrogen ammonia for the treated group, suggesting that essential oils could not modulate deamination nor the population of ammonia producing bacteria [47]. One of the characteristics of the essential oils is modulated ruminal microbiota and, consequently, fermentation and nutrient degradation in the forestomach [18, 55].

For all ruminal parameters, a week effect during preweaning was observed ($P \leq 0.05$, Table 4). Those findings were expected since ruminal parameters are related to increased starter intake, rumen development, microbiota colonization, and calf development to become a ruminant [4].

## Blood cell count, metabolites and IGF-1

During the pre- and post-weaning periods, all blood metabolites were not altered by BEO supplementation (Table 5). Similar patterns of BHB, glucose [35, 47], urea [37], total plasma

**Table 5. Pre- and post-weaning mean blood concentrations of insulin growth factor type 1 (IGF-1) and metabolites of control heifers (CON) and heifers supplemented with a blend of essential oils blend (BEO) in milk replacer during pre-weaning.**

| Item | Treatment | | SEM | P–value[3] | | |
|---|---|---|---|---|---|---|
| | CON[1] (n = 15) | BEO[2] (n = 14) | | T | W | T x W |
| **Pre-weaning (4 to 60 d)** | | | | | | |
| BHB (mmol/L) | 0.17 | 0.12 | 0.02 | 0.43 | 0.001 | 0.98 |
| Urea (mg/dL) | 24.55 | 22.69 | 3.76 | 0.16 | 0.02 | 0.31 |
| Glucose (mg/dL) | 100.35 | 102.97 | 16.50 | 0.49 | 0.15 | 0.56 |
| IGF-1 (ng/mL) | 101.95 | 93.16 | 32.4 | 0.38 | <0.001 | 0.27 |
| **Post-weaning (61 to 90 d)** | | | | | | |
| BHB (mmol/L) | 0.36 | 0.37 | 0.10 | 0.70 | <0.001 | 0.13 |
| Urea (mg/dL) | 24.57 | 22.73 | 4.34 | 0.16 | 0.01 | 0.34 |
| Glucose (mg/dL) | 88.45 | 84.74 | 8.65 | 0.29 | 0.22 | 0.14 |
| IGF-1 (ng/mL) | 160.70 | 175.94 | 23.4 | 0.43 | 0.31 | 0.12 |

[1]CON = control

[2]BEO = 1 g/calf/d blend of essential oil.

[3]T = treatment effect; W = week effect, T x W = treatment by week interactions.

protein, and IGF-1 [56] were found in both treatments. Nevertheless, BHB and urea increased with age ($P \leq 0.05$, Table 5), since they are directly correlated with fatty acid metabolism and ruminal ammonia concentration, respectively [57]. The IGF-1 concentration increased with age on the preweaning phase ($P \leq 0.001$). Since this hormone is a mitogen and related to cell proliferation and differentiation, it is correlated with BW and animal growth [58].

Glucose did not change during the pre-weaning phase and decreased during the post-weaning period (Table 5). Taking into account that calves use glucose as a primary source of energy in the firsts weeks of age, these age-related changes are associated with changes in diet and rumen development [59]. After weaning, calves complete their rumen development and, VFA produced by ruminal microbiota becomes the primary energy source, justifying BHB concentration increase, and glucose concentration decrease [5, 60]. However, since there were changes in C2:C3 proportion in the BEO, the increase of propionic acid could consequently impact glucose blood concentration. Since essential oils can increase insulin sensitivity, not finding glucose differences between treatments does not mean that there were no changes in the glucose pathway [38, 39]. Therefore, further investigations over these aspects are needed.

All blood cell counts were within normal range based on age and species normality. Changes in blood cell count are typical during heifer growth, and blood cells tend to increase with animal age [61]. These changes corroborate with the week effect on mean corpuscular volume (MCV), basophils, eosinophils, segmented neutrophils, lymphocytes, monocytes, and platelets ($P = 0.04$). There were no differences in erythrogram parameters between BEO and CON (Table 6). Leukogram parameters showed decreased counts of basophil and platelet cells in BEO treatment ($P \leq 0.05$). Basophils and platelets originate from different myeloid

**Table 6. Pre-weaning hematological parameters of control heifers (CON) and heifers supplemented with a blend of essential oils blend (BEO) in milk replacer during pre-weaning.**

| Item[1] | Treatment | | SEM | P–value[4] | | |
|---|---|---|---|---|---|---|
| | CON[2] (n = 15) | BEO[3] (n = 14) | | T | W | T x W |
| RBC (x 10^6/μL) | 8.02 | 7.95 | 0.88 | 0.86 | 0.63 | 0.87 |
| PCV (%) | 35.53 | 35.21 | 5.05 | 0.85 | 0.11 | 0.69 |
| Hb (q/dL) | 11.07 | 10.94 | 1.61 | 0.81 | 0.14 | 0.73 |
| MCV (fL) | 44.74 | 44.51 | 2.94 | 0.74 | <0.001 | 0.51 |
| MCHC (%) | 31.10 | 31.14 | 0.76 | 0.87 | 0.15 | 0.99 |
| Total leukocytes (/μL) | 10,908.45 | 11,200.78 | 2,630.0 | 0.76 | 0.19 | 0.22 |
| Basophils (/μL) | 2.14 | 0.00 | 1.03 | <0.001 | <0.001 | <0.001 |
| Eosinophils (/μL) | 68.40 | 143.90 | 0.66 | 0.24 | <0.001 | 0.36 |
| Band neutrophil (/μL) | 31.76 | 26.22 | 5.69 | 0.68 | 0.83 | 0.31 |
| Segmented neutrophils (/μL) | 5,300.63 | 5,286.56 | 1,700.0 | 0.98 | <0.001 | 0.78 |
| Lymphocytes (/μL) | 4,837.40 | 5,082.82 | 1,120.0 | 0.66 | <0.001 | 0.01 |
| Monocytes (/μL) | 421.60 | 466.00 | 247.0 | 0.48 | 0.01 | 0.29 |
| Platelet (x 10^3/μL) | 410.41 | 353.70 | 108.0 | 0.04 | <0.001 | 0.10 |
| Plasmatic protein (g/dL) | 6.03 | 6.03 | 0.72 | 1.00 | 0.17 | 0.40 |
| PLR | 0.08 | 0.08 | 0.03 | 0.91 | 0.02 | 0.04 |
| NLR | 1.26 | 1.46 | 0.03 | 0.60 | <0.001 | 0.55 |

[1]RBC: red blood cell, PCV: packed cell volume, Hb: hemoglobin, MCV: mean corpuscular volume, MCHC: mean corpuscular hemoglobin concentration, PLR: platelet lymphocyte ratio, NLR: neutrophils lymphocytes ratio.

[2]CON = control

[3]BEO = 1 g/calf/d blend of essential oil.

[4]T = treatment effect; W = week effect; T x W = treatment by week interactions.

precursors and, both play essential roles in inflammation balance and immune response development in mammal [62]. The lower counts of basophil and platelets on BEO treatment may influence and modulate inflammatory response by secretion of immune modulators [63], growth factors, or chemotaxis on a variety of white blood cells [43]. This modulation could help explain an interaction effect found for lymphocytes (Fig 2), where values of d 30 and 60 were different from d 1 with an accentuated increase in BEO. There have been reports of immune response potentiation of piglets supplemented with essential oils. The animals had improved lymphocyte proliferation, phagocytosis rate, and humoral immune response [54].

Oregano and thyme oils supplemented to Holstein calves positively influenced erythrogram parameters, lymphocytes, neutrophils, and band neutrophils with higher values for treated calves [64]. For older animals, it has been shown a linear increase in the values for lymphocyte and monocyte counts for heifers supplemented with plant extract containing essential oils [65]. Hence, agents with antioxidant activity, like essential oils, can reduce platelet activation and consequently reduce oxidative stress and inflammation [66]. Platelets also play a central role in the coagulation process. Different essential oils have been used for thrombosis treatment in humans, acting on platelet aggregation and its thromboxane synthesis [67]. Although our results demonstrate a decrease in basophil and platelet counts, it is necessary to perform novel experiments to characterize the effects of BEO on the inflammatory and coagulation process in heifers. Differences between PLR and NLR were not found (Table 7). These ratios are inflammatory markers and inform disease activity, being a useful tool to understand inflammation pathophysiology and immune response [68].

## Health measurements and minimum inhibitory concentration

Diarrhea is the most prevalent disease for calves under one month of age. Causes for juvenile diarrhea include a combination of factors but are generally related to viral, bacterial, or/and protozoa infection [2]. Coronavirus, rotavirus, *Salmonella* spp. and/or *Cryptosporidium parvum* are the most common agents under 14 d of age. *Salmonella* spp., *Eimeria* spp. and/or *Giardia* spp. are the most common pathogens in older calves [2, 69].

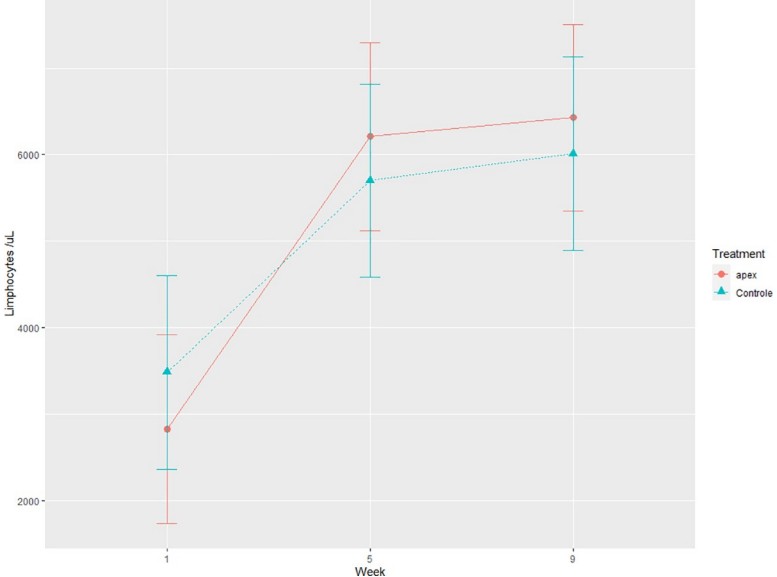

**Fig 2. Lymphocytes values of control heifers (CON) and heifers supplemented with 1.0 g/calf/ d of a blend of essential oils (BEO) in milk replacer during the pre-weaning period.**

**Table 7. Pre and post-weaning mean values of the fecal score, respiratory score, days with a respiratory score above 4, days with fever, days with diarrhea, days with severe diarrhea of control heifers (CON) and heifers supplemented with a blend of essential oils (BEO) in milk replacer during pre-weaning.**

| Item | Treatment | | SEM | P–value[3] | | |
|---|---|---|---|---|---|---|
| | CON[1] (n = 15) | BEO[2] (n = 14) | | T | W | T x W |
| **Pre-weaning (4 to 60 d)** | | | | | | |
| Fecal score[4] | 0.54 | 0.45 | 0.04 | 0.04 | <0.001 | 0.18 |
| Respiratory score[4] | 0.79 | 0.69 | 0.02 | 0.22 | <0.001 | 0.02 |
| Days with respiratory score > 4[5] | 0.00 | 0.14 | 0.05 | 0.44 | – | – |
| Days with fever | 0.94 | 0.98 | 0.20 | 0.66 | – | – |
| Days with diarrhea | 7.87 | 5.79 | 0.71 | 0.24 | – | – |
| Days with severe diarrhea | 3.13 | 1.93 | 0.37 | 0.12 | – | – |
| **Post-weaning (61 to 90 d)** | | | | | | |
| Fecal score | 0.04 | 0.04 | 0.009 | 0.43 | 0.68 | 0.95 |
| Respiratory score | 1.10 | 1.03 | 0.05 | 0.59 | <0.001 | 0.74 |
| Days with respiratory score > 4 | 0.00 | 0.00 | – | – | – | – |
| Days with fever | 0.52 | 0.90 | 0.23 | 0.21 | – | – |

[1] CON = control

[2] BEO = 1 g/calf/d blend of essential oil.

[3] T = treatment effect; W = week effect, T x W = treatment by week interactions.

[4] Scores were adapted to follow the University of Wisconsin calf health scoring chart [2].

[5] There were no days with respiratory score > 4 during the post-weaning period.

The supplementation of essential oils has already shown beneficial results for lowering diarrhea and fecal scores in other species with the same efficiency of AGPs [18, 31, 70]. For piglets, where this is a prevalent disease and caused by similar agents as in calves, it has been shown favorable results with lower diarrhea prevalence for treated animals [70]. In our study, the average age for diarrhea (scores 2 and 3) occurrence was 12.2 ± 3.6 d for BEO and 13.6 ± 3.8 d for CON with no statistical difference (P = 0.54). Diarrhea incidence on pre-weaning in BEO treatment was 85% against 93% for CON treatment with no statistical difference (P = 0.68). The fecal score was different between treatments (P = 0.04), with lower values for BEO, and changed over time (P ≤ 0.001, Table 7). Days with diarrhea (scores 2 and 3, P = 0.24) and days with severe diarrhea (score 3, P = 0.12) were not different between treatments (Table 7). Three animals of each treatment were medicated for diarrhea with anti-inflammatories, and the therapy duration was 1.6 ± 0.57 d for BEO and 3.0 ± 1 d for CON. It is noteworthy that this treatment was done outside the hemogram and total cell count evaluation in this study. Besides no differences in the diarrhea prevalence, the lower fecal score in the BEO could point to better gut health and less microbiota disability [54]. However, is important to point out that we did not collect samples to analyze microbiota changes before, during, and after diarrhea, or pathogenic bacteria count in feces.

Evaluation of the respiratory score parameters indicated that 2 BEO animals and 1 of CON animals exceeded score 4, indicating respiratory disease on pre-weaning. The average days with a high score were 1.0 ± 0 d for BEO and CON. No effect was found on days with high respiratory score or number of affected animals. However, a week and an interaction week x treatment effect on pre-weaning was observed, with the difference between treatment scores and lower values for the BEO in week 2 (P = 0.02, Table 7, Fig 1). The second week was the period in which animals had a higher incidence of diarrhea. It is known that diarrhea and respiratory problems are caused by a combination of factors and related to the immunity status, nutrition, type of housing, and season [2]. Herds with respiratory diseases in calves have

more diarrheal disease [71]. Thus, in this trial, the respiratory signs could be related to the previous enteric disease. Weeks 5 and 6 showed a lower score difference between treatments and a lower incidence of respiratory signs. The number of treated animals was 2 for BEO only during the preweaning period, with an average of treatment days of 1.3 ± 1.4, and 3 for CON with an average of treatment days of 2.0 ± 0.57. Treatments occurred only in the pre-weaning period using antibiotics and anti-inflammatories.

Pneumonia is usually associated with the post-weaning phase. However, it may affect younger calves [2]. Post-weaning respiratory scores revealed higher mean values when compared with pre-weaning, but no animals had scores above 4. There was a week effect ($P \leq 0.001$), in week 12, probably due to weaning and dehorning stress.

It has been reported that essential oils have an antiseptic and antimicrobial activity that may help balance intestinal microbiota [72]. Gram-positive bacteria are the most sensitive to the essential oils microbial activity [18, 23], but Gram-negative bacteria and some types of parasites can also be susceptible [16] to different essential oils. Thus, some essential oils could reduce the incidence and severity of diarrhea syndrome in calves through inhibition of coliform overgrowth [73]. The in vitro test with BEO in 1.0 μg/mL concentration did not inhibit bacterial growth–both *E. coli* and *S. Typhimurium*. Thus, at this concentration, BEO did not have any direct antibacterial effect. However, besides no direct influence found over the bacterial evaluation, BEO calves presented differences on basophil (Table 6) and lymphocyte cell populations (Fig 2), which could be associated with modulation of the inflammatory immune response. Thus, outcomes found on fecal and respiratory scores could be related to indirect changes in hemato-biochemical parameters and not with a direct antibacterial effect.

## Conclusions

Feeding BEO to pre-weaned heifers on MR did not affect intake, performance parameters, blood metabolites, or IGF-1 concentration. However, it changed C2:C3 proportion during pre- and post-weaning periods, showed signs of immunity improvement, and lower fecal scores in the pre-weaning phase. Therefore, essential oils are a health additive option to modern production systems and could be used as an alternative to improve calf health and performance. Further research is needed to define the best route and dosage, understand the contribution of essential oils to decrease neonatal diseases' morbidity, and verify the possible interaction with other molecules.

## Supporting information

**S1 Data.**
(XLSX)

## Author Contributions

**Conceptualization:** Sandra Gesteira Coelho, Alessandra L. Voorsluys, David V. Jacob, Mariana Magalhães Campos.

**Data curation:** Joana Palhares Campolina, Anna Luiza Belli.

**Formal analysis:** Joana Palhares Campolina.

**Funding acquisition:** Sandra Gesteira Coelho, Mariana Magalhães Campos.

**Investigation:** Joana Palhares Campolina, Anna Luiza Belli, Rodrigo Otávio S. Silva.

**Methodology:** Joana Palhares Campolina, Sandra Gesteira Coelho, Anna Luiza Belli, Fernanda Samarini Machado, Luiz Gustavo R. Pereira, Mariana Magalhães Campos.

**Project administration:** Sandra Gesteira Coelho, Fernanda Samarini Machado, Mariana Magalhães Campos.

**Resources:** Sandra Gesteira Coelho, Fernanda Samarini Machado, Luiz Gustavo R. Pereira, Thierry R. Tomich, Mariana Magalhães Campos.

**Software:** Joana Palhares Campolina.

**Supervision:** Sandra Gesteira Coelho, Fernanda Samarini Machado, Luiz Gustavo R. Pereira, Thierry R. Tomich, Mariana Magalhães Campos.

**Validation:** Sandra Gesteira Coelho, Mariana Magalhães Campos.

**Visualization:** Sandra Gesteira Coelho.

**Writing – original draft:** Joana Palhares Campolina.

**Writing – review & editing:** Joana Palhares Campolina, Sandra Gesteira Coelho, Anna Luiza Belli, Fernanda Samarini Machado, Luiz Gustavo R. Pereira, Thierry R. Tomich, Wanessa A. Carvalho, Rodrigo Otávio S. Silva, Alessandra L. Voorsluys, David V. Jacob, Mariana Magalhães Campos.

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
