## [Decision Letter · Decision Letter 0]

17 Jun 2020

PONE-D-20-07435

Effects of a blend of essential oils in milk replacer on performance, rumen fermentation, blood parameters and health scores of dairy heifers

PLOS ONE

Dear Dr. Palhares Campolina,

Thank you for submitting your manuscript to PLOS ONE. After careful consideration, we feel that it has merit but does not fully meet PLOS ONE’s publication criteria as it currently stands. Therefore, we invite you to submit a revised version of the manuscript that addresses the points raised during the review process.

We look forward to receiving your revised manuscript.

Kind regards,

Juan J Loor

Academic Editor

PLOS ONE

Journal Requirements:

The authors thank Professor Armando Cunha Jr. for helping perform MIC analyzes,

Professor Ângela Quintão, Professor Fabiola Paes Leme and Vera Carsoso Ferreira

Aiken for helping on this paper. We also thank Coordenação de Aperfeiçoamento de

Pessoal de Nível Superior (CAPES, Brasília, Brazil), Fundação de Amparo à Pesquisa

do Estado de Minas Gerais (FAPEMIG, Minas Gerais, Brazil), Conselho Nacional de

Desenvolvimento Científico e Tecnológico (CNPq, Brasília, Brazil), Instituto Nacional

Ciência e Tecnologia Ciência Animal (INCT, Viçosa, 598 Brazil), Embrapa Dairy Cattle

(Minas Gerais, Brazil) and Adisseo Company for financial support of this research.

The funders had no role in study design, data collection and analysis, decision to

publish, or preparation of the manuscript.

The authors have declared that no competing interests exist.

We note that one or more of the authors are employed by a commercial company: Adisseo, Campinas.

Please respond by return email with an updated Funding Statement and Competing Interests Statement and we will change the online submission form on your behalf.

Reviewers' comments:

Reviewer's Responses to Questions

**Comments to the Author**

1. Is the manuscript technically sound, and do the data support the conclusions?

Reviewer #1: Partly

Reviewer #2: Yes

2. Has the statistical analysis been performed appropriately and rigorously? 

Reviewer #1: Yes

Reviewer #2: Yes

3. Have the authors made all data underlying the findings in their manuscript fully available?

Reviewer #1: Yes

Reviewer #2: Yes

4. Is the manuscript presented in an intelligible fashion and written in standard English?

Reviewer #1: No

Reviewer #2: Yes

5. Review Comments to the Author

Reviewer #1: Overall, the topic is relevant and the objective stood to contribute to the exploration of bioactive ingredients as an alternative to antimicrobials. However, the presentation of the results and discussion lacked substance and did not accurately convey outcome or inferences for future use. Erring on benefit of the doubt with respect to translation difficulties; please extensively review for spelling, grammar and context. It was especially hard to follow logic and connection between referenced work and the results observed in the paper during the discussion. The authors have done good research but significant and major English editing and organization is needed. I have made some recommendations for edits but do not consider it an exhaustive list of what could be changed.

Are there any references from swine or poultry that could be used throughout this paper that would integrate into the concepts you are trying to address? It is a fairly new area of interest in calves but here is a more prevalent publications in these other species.

Inconsistency with how references are cited within manuscript. PLoS one says to use reference numbers, but the use of author names and numbers varies throughout. If able to streamline without compromising word flow, please do so.

Do not start sentences with abbreviations.

Italicize “P” values throughout

Line 43, 115, 137-138, 282-284: you indicate that this is a complete block design but you enrolled different numbers of animals between the two treatments. Can you explain this? Were any animals enrolled and not used in the final dataset?

Title

Line 3: add “,” after “blood parameters”

Abstract

Line 28: “on” should be “during”

Line 33: insert a space between with1

Is commercial name needed here? May be more beneficial to list oils included in mixture?

Was BEO supplied in MR?

Line 34: should be reconstituted to 15% not “at”

Line 37: Delete “The outcomes”

Line 42: “each” should be every. You have 14 days listed here but that is different from what is described in the materials and methods. Please rectify.

Line 46-48: The terminology used here saying you accept the alternative hypothesis is inappropriate. Either fail to reject or reject the null.

Line 51: Tense switch from past to present; change “is” to “was” after “effect”.

Line 51-55: italicize P values. Are P-values in abstract allowed in Plos One?

Line 53: what does long-term mean? How does your data support long term immunological effects?

Introduction

Lines 60-68: This paragraph does not really add to the justification for the objectives addressed in this paper; consider starting introduction at Line 69 instead.

Line 70: insert a “,” after mortality rates

Line 71-72: Consider rewording to make more sense chronologically: “Therefore, tools that improve calf development and health are essential to reduce disease, mortality and morbidity as well as accelerate heifer development.”

Line 73: insert “as a” between functionally and non-ruminant, delete “it must have its” and replace with “the”, the rest of that sentence needs to be reworded for clarity.

Line 77: Remove “anymore”.

Line 78: Remove “Since” Is there a reference(s) for this statement?

Line 85-87: Remove entire sentence starting with “Furthermore…”. I think this statement is not based on fact but opinion. There are no differences between the health benefits of cows managed under an organic system vs a system that utilizes antibiotics. Are there references for these statements?

The entire paragraph (lines 78-88) has the potential to be a slippery slope discussion regarding antibiotics so be careful with what you choose to include and how you make relation to natural alternatives. I think it could be framed in a much different context to discuss examples like the Veterinary Feed Directive in the US or limits on antibiotics as growth promoters are being phased out if not already illegal.

Line 84: Change “over” to “on”.

Line 85-87: Mention of organic dairy farms is extraneous and should be removed. This statement has no bearing on overall objective of this paper.

Line 91: change “on” to “with”

Line 93: change “for” to “to”

Line 94: ‘Leucocyte” should be “leukocyte”

Line 95: Suggest “Lastly, essential oils have been shown to function similarly to ionophores by influencing…”

Line 100: change “over” to “on”

Line 106: the word ‘influence’ alone is unclear; was positive or negative influence hypothesized?

Materials and Methods

Line 120-121: ‘up to’ should be ‘before’

Line 122: “At 2 to 3 d of age” should be “From 2 to 3 d of age” Correct? You fed transition milk on d 2 and 3?

Line 125, 200: the g should be italicized also is this g force value correct? I think it is rpm. It is different than other locations throughout materials and methods. Please verify.

Line 127: you state that only heifers with a high enough Brix value were enrolled. Were there any heifers that were not enrolled because of this criteria?

Line 129-130: was intake recorded of water and starter? When were these first offered?

Line 132: recommend “On d 4 of age, heifers were assigned to one of two experimental milk replacers. Both milk replacers were fed at a rate of 5 L/d…” May be able to reorganize and consolidate text you have on line 135. Feeding rate could be described after description of treatment milk replacers

Line 135: replace “with” with “at”

Line 136- you state serum total protein but you actually measured Brix correct?

Line 137: How was dosage of 1g/calf/d determined? Is it manufacturer recommendation? Suggest addition of “or a commercial blend of essential oils supplemented at a rate of 1 g/calf/d (BEO; Apex..)

Line 138: Delete “Apex Calf” from the second half of that line. Recommend “The BEO is a dry powder…”

Line 140: essential oil “for” each meal

Line 141: milk or milk replacer? Were calves fed the control diet fed and treated in a similar way or did they receive their whole meal at one time? Did this influence timing of feeding between treatments that might influence blood measurements collected relative to feeding?

Table 1: Is this analyzed composition? Table heading should stand alone, suggest addition of “fed to calves”. You said you took samples for nutrient analysis but did not present standard deviations of feed ingredients. “unless otherwise noted” can be deleted. Extra space between % of in Organic matter heading. Was starch measured? It would be a good addition to analysis. Superscript 1 is not required. Superscript 2- if commercial milk can you provide brand name and company? Starter ingredients I think it is useful to know but not sure if entire list is necessary in this case.

Line 153 and 158: starter is listed to have monensin and a probiotic additive. How does this impact your treatments? Any consideration for this when interpreting your results? In intro you discuss ionophores… but do not say anything further.

Line 180: inconsistent use of “body development” and “structural growth”. I think structural growth more appropriate for the measures you collected

Line 181: “birth date” should be “day of birth”. BW measurements every 3 d needs a bit more clarification. How was this handled for analysis? Did the day BW was collected continually change day of week?

Line 192: Introduction states that rumen fluid was collected every 2 wk; however, collection days listed do not follow a 14-day pattern. Days listed are 14, 28, 42, 60, 74, and 90; a 14-day pattern would be 14, 28, 42, 56, 70 and 84. If samples were collected in a different timeframe than originally stated, please discuss. It seems as though the later collection dates were adjusted to account for collection on days that calves were weaned (60d) and also at the end of the collection period (90).

Line 198: spelling error; ‘termo’ should be “Thermo”. Add “WI” after “Madison.”

Line 206- insert “3 h after morning feeding” after collected

Line 207: “feed” should be “ingestion”

Blood samples not collected in weekly increments as stated in abstract; collection days after baseline collection at day 4 follow 7 day pattern until d 63, then are collected 11, 7 and 9 d later instead of consistently 7. Correct throughout to represent what was actually done.

Line 209: Collection days again do not match up to the initial 2-week timeframe stated for IGF-1 determination. Fluctuates between 7 and 14 day collection. Correct this throughout

Line 222: essay should be assay

Line 227: perform should be evaluate or analyze

Line 235: delete “it was also performed” and add “were calculated” at the end of the sentence.

Line 244: here you indicate classification of severe diarrhea but terminology used in results and discussion is different so could you clarify between the two sections of text what “severity” of diarrhea you are discussing please?

Line 255: “Final respiratory score considered the sum of all punctuations” should be reworded to “A final respiratory score was determined by the summation of all scores”. However could you clarify if diarrhea score was included in this or not? Maybe list which scores when into the final sum?

Line 257: What antibiotic was used, what dosage, how long?

Line 258 and 261: “subsequently” should be sequential

Line 260: What is ‘pulmonary commitment’?

Line 263: Consider moving this section to earlier in materials and methods section and discuss more about why it was used. Was this used to determine dosage?

Line 267: No footnote number after author reference; inconsistent with remainder of manuscript and instructions to authors

Line 269: is the tween from a specific company?

Line 290: Weren’t all the animals of similar genetic composition? Why was this used as a blocking effect?

Line 294-295: Rephrase “accept or deny”

Line 297-298: were mean values that were transformed presented as back transformed values? SE would generally not be presented with transformed values. It would be a confidence interval

Results and discussion

Line 307: Remove ‘s’ from ‘first’.

First paragraph or in R and D: what about timing of major enteric or other diseases that calves experience and timing of occurrence? Thinking about diarrhea in calves occurring mainly in first 3 wk of life and limited starter intake during this time which would support inclusion in milk if to benefit this situation. What about timing and main use of antibiotics relative to these health events? That is a main motivation for use of essential oils correct?

Line 311-312: how was this determined? Not described

Line 316: Total mixed ration instead of total mixed ratio

Line 319: consider changing ‘supply’ to ‘delivery method’ to be more clear

Line 332-333: there is no description of a correlation in stats section. Please include in Stats section

Line 333: spelling error; ‘reveled’ should be ‘revealed’

334-335: impacted on smaller should be resulted in reduced

Line 344: EO was not defined previously and this sentence was already stated in the second paragraph of this section. Reorganize and consolidate

349-350: this sentence does not make sense

Line 352: Table 3 should be placed here to remain consistent with rest of document and journal requirements

Line 356: Italicize P

Table 3: your SEM seem like it would result in a difference between treatments for BW at weaning and final BW are you underpowered to be able to detect these differences?

ADG should be kg/d and feed efficiency should be kg/kg

Any initial measure of body weight or structure used for covariate for respective variable?

Line 373: ‘present’ should be ‘presented’

Line 376: how was “heifers’ ingestion behavior” evaluated?

Line 379: spaces between numbered citations

Line 388: reference for this statement?

Line 395: “Besides the previous cited effects” is vague. Define a little more specifically in terms of health benefits, etc.

Line 397: Simply saying that essential oils can cause a toxic effect could insinuate a negative connotation to animal health; reword to relate better to its detriment on harmful bacteria rather than to the rumen, which is how it reads as currently written.

Line 399: Similar thoughts on use of the word ‘consequence’.

Line 409-410: rewrite this section it is really confusing. Group without EO would be CON correct?

Line 410-415: what about positive benefits to small intestine on whole GIT development? There is a little bit of information on this or theories at least.

Line 415: Not sure of use of extravasation; considering its definition it doesn’t fit how you’re using it

Line 419-428: This paragraph does not contribute a lot to the overall results; very weak correlations between references and findings. Context is needed from other references cited to add more meaning to the discussion. Not sure it fits here

Line 458: remove ‘of’ before ‘insulin’. What are the references for these conclusions?

Line 460: remove ‘s’ from ‘these’.

Table 6. define PLR and NLR

Line 503-506: were these measured in your study? Connection to your results?

Line 511: and/or

Line 523: what cell count?

Line 525: health score is respiratory score?

Table 7: when was temperature or health evaluated daily? Could you add this to materials and methods

I don’t think full score explanation is need again. Reference McGuirk

Line 554-556: Absolutely cannot draw a correlation between or make an assumption about respiratory signs and diarrhea.

Line 571: spelling error: ‘trough’ should be ‘through’

Line 566-579: Should include potential effects due to monensin and probiotic additives in starter on results here. If you did not see an in vitro effect at the concentration of 1.0 µg/mL, would that not suggest that you consider a different dosage?

Line 586- rout should be route

Figure 1 has labels that are cutoff, need units for variables in stead of “value”, “Trat” should be treatment

Figure 2 lymphocytes is spelled incorrectly

Reviewer #2: The authors present in the manuscript an exciting and original research idea relevant to the performance of dairy calves, and reserves to be published.

In general, the manuscript is well written, the statistical analyses are appropriate, and different parts are well presented and explained. After careful review of the document, the reviewer has the following minor suggestions:

ABSTRACT

Consider reporting the specific P-value instead of a general P ≤ 0.001 or P ≤ 0.05.

MATERIALS AND METHODS

Line 140: essential oils instead of essential oil.

Line 293: define performance. Does it mean BW and body measures?

RESULTS AND DISCUSSION

Similar to the abstract section, consider reporting the specific P-value in the text instead of a general P ≤ 0.001 or P ≤ 0.05.

Line 334: inverse association…, were. Did you mean where?

Line 341 – 351: consider adding of discussing why the lack of difference in responses in your study compared with those cited.

The reviewer considers that the authors should present the fatty acid composition or profile, at least the major fatty acids present in the BEO. It is a fundamental analysis to include in the manuscript and accounts for in the results and discussion section.

6. PLOS authors have the option to publish the peer review history of their article (what does this mean?). If published, this will include your full peer review and any attached files.

Reviewer #1: No

Reviewer #2: No

---

## [Author Response · Author response to Decision Letter 0]

14 Aug 2020

Dr. Juan Loor #1: Please ensure that your manuscript meets PLOS ONE's style requirements, including those for file naming. The PLOS ONE style templates can be found at https://journals.plos.org/plosone/s/file?id=wjVg/PLOSOne_formatting_sample_main_body.pdf and

Au: Thank you for your contribution. We rechecked all style requirements to meet PlosOne formatting.

Dr. Juan Loor #2: Thank you for stating the following in the Acknowledgments Section of your manuscript:

The authors thank Professor Armando Cunha Jr. for helping perform MIC analyzes, Professor Ângela Quintão, Professor Fabiola Paes Leme and Vera Carsoso Ferreira Aiken for helping on this paper. We also thank Coordenação de Aperfeiçoamento de Pessoal de Nível Superior (CAPES, Brasília, Brazil), Fundação de Amparo à Pesquisa do Estado de Minas Gerais (FAPEMIG, Minas Gerais, Brazil), Conselho Nacional de Desenvolvimento Científico e Tecnológico (CNPq, Brasília, Brazil), Instituto Nacional Ciência e Tecnologia Ciência Animal (INCT, Viçosa, 598 Brazil), Embrapa Dairy Cattle

(Minas Gerais, Brazil) and Adisseo Company for financial support of this research.

Au: Thank you for your point. We have taken away this acknowledgment section, and we will update the funding part.

Dr. Juan Loor #3: Thank you for stating the following in the Competing Interests section:

The authors have declared that no competing interests exist.

We note that one or more of the authors are employed by a commercial company: Adisseo, Campinas. 

Please also include the following statement within your amended Funding Statement. “The funder provided support in the form of salaries for authors [insert relevant initials], but did not have any additional role in the study design, data collection and analysis, decision to publish, or preparation of the manuscript. The specific roles of these authors are articulated in the ‘author contributions’ section.”

Please respond by return email with an updated Funding Statement and Competing Interests Statement and we will change the online submission form on your behalf.

Au: Thank you for your concern. The company organization did not play a role in the study design, data collection and analysis, decision to publish, or preparation of the manuscript and only provided financial support in the form of research materials. They also do not have any decision on the sharing of data and materials. The two authors that are employed by a commercial company helped with providing materials for the trial on the revision to publish the article.

Reviewer #1: Overall, the topic is relevant, and the objective stood to contribute to the exploration of bioactive ingredients as an alternative to antimicrobials. However, the presentation of the results and discussion lacked substance and did not accurately convey outcome or inferences for future use. Erring on benefit of the doubt with respect to translation difficulties; please extensively review for spelling, grammar and context. It was especially hard to follow logic and connection between referenced work and the results observed in the paper during the discussion. The authors have done good research but significant and major English editing and organization is needed. I have made some recommendations for edits but do not consider it an exhaustive list of what could be changed.

Au: Thank you for your concern. We have made an extensive English re-editing and reviewing for spelling, grammar, and organizing the ideas to fit the context. We reorganized sentences and added new information to enrich the discussion.

Are there any references from swine or poultry that could be used throughout this paper that would integrate into the concepts you are trying to address? It is a fairly new area of interest in calves but here is a more prevalent publications in these other species.

Au: Thank you for your concern. We added information concerning other species, as well as new releases in calves’ studies supplemented with essential oils

Inconsistency with how references are cited within manuscript. PLoS one says to use reference numbers, but the use of author names and numbers varies throughout. If able to streamline without compromising word flow, please do so.

Au: Thank you for your concern. Edited. Although PloS One accepts author’s name in the text, we agree that taking them out helped with word flow. 

Do not start sentences with abbreviations.

Italicize “P” values throughout

Au: Edited. Thank you.

Line 43, 115, 137-138, 282-284: you indicate that this is a complete block design but you enrolled different numbers of animals between the two treatments. Can you explain this? Were any animals enrolled and not used in the final dataset?

Au: Thank you for your contribution. Initially, we enrolled 15 animals in each treatment, with a total of 30 animals. However, one heifer form BEO treatment was not growing and had struggled with innumerous wealth problems. She was then euthanized, and it was discovered that she had a malformation of her intestinal tract. Because of that matter, she was taken out of the study. A new heifer was not enrolled in her place since the trial was already going on, and the animals were at least 30 days old. Enrolling a new animal after that long could change animal behave and performance due to contact with older animals. We also started the trial in March, which is fall in Brazil. If we decided to enroll a new animal, she would be born in winter. For those reasons, we decided then to have an unbalanced design. (L44, 128, 152-154, 302-304).

Title

Line 3: add “,” after “blood parameters”

Au: Included, thank you. (L3).

Abstract

Line 28: “on” should be “during”

Au: Edited, thank you. (L28).

Line 33: insert a space between with1

Au: Included, thank you. (L32).

Is commercial name needed here? May be more beneficial to list oils included in mixture?

Au: Thank you for your contribution. We choose to put the commercial name of the used product since some papers do essential oil extract and can have different results. We do have which extracts plants compose the product. However, we think if we put which product we used, it can give more information to the reader. However, also added the plant extracts that component to the abstract: “composed by plant extracts derived from anise, cinnamon, garlic, rosemary and thyme” (L33-34).

Was BEO supplied in MR?

Au: Thank you for your contribution. On line 24, we said that the BEO was supplied in MR. However, to help a better understanding of the text, we also added at L32.

Line 34: should be reconstituted to 15% not “at”

Au: Edited, thank you. (L35)

Line 37: Delete “The outcomes”

Au: Edited, thank you. (L38).

Line 42: “each” should be every. You have 14 days listed here but that is different from what is described in the materials and methods. Please rectify.

Au: Edited, thank you (L43).

Line 46-48: The terminology used here saying you accept the alternative hypothesis is inappropriate. Either fail to reject or reject the null.

Au: Thank you for your concern. We edited to hypothesis to a more appropriate terminology (L48-50).

Line 51: Tense switch from past to present; change “is” to “was” after “effect”.

Au: Edited, thank you (L52)

Line 51-55: italicize P values. Are P-values in abstract allowed in Plos One?

Au: Edited, thank you (L52-54, 56). We also check on PlosOne submission rules, and they do not say about allowing it or not P-values, but they said the abstract should: “Summarize the most important results and their significance.” We also found other papers already publish in PlosOne that used P-values in the abstract: Pone.0191687 and Pone.0146488. Therefore, we thought that it would be interesting to leave the P-values (L 52-55).

Line 53: what does long-term mean? How does your data support long term immunological effects?

Au: We apologize for the misunderstanding. We reformulated the sentence to make clear the statements (L54-55). 

Introduction

 Lines 60-68: This paragraph does not really add to the justification for the objectives addressed in this paper; consider starting introduction at Line 69 instead.

Au: Thank you for your contribution. The manly parameters evaluated in a good calf rearing program are average dairy gain and disease incidence. Since we also did in this trial, we thought that it would be nice to explain why they are so important and why we choose to measure them. It is also important to point that we did measure immune system parameters, and those are highly correlated with weight gain and animal’s health. For that reason, we decided to keep this paragraph (L61-70)

Line 70: insert a “,” after mortality rates

Au: Included, thank you (L33)

Line 71-72: Consider rewording to make more sense chronologically: “Therefore, tools that improve calf development and health are essential to reduce disease, mortality and morbidity as well as accelerate heifer development.”

Au: We apologize for the misunderstanding. We reformulated the sentence to make clear the statement (L72-73).

Line 73: insert “as a” between functionally and non-ruminant, delete “it must have its” and replace with “the”, the rest of that sentence needs to be reworded for clarity.

Au: Thank you for your contribution. We reformulated the whole sentence (L73-75).

Line 77: Remove “anymore”.

Au: Edited, thank you (L77)

Line 78: Remove “Since” Is there a reference(s) for this statement?

Au: Edited, thank you. The reference for that statement comes in the next sentence (Drackely, 2008). However, we added at the end of the sentence to be precise (L70).

Line 85-87: Remove entire sentence starting with “Furthermore…”. I think this statement is not based on fact but opinion. There are no differences between the health benefits of cows managed under an organic system vs a system that utilizes antibiotics. Are there references for these statements?

The entire paragraph (lines 78-88) has the potential to be a slippery slope discussion regarding antibiotics so be careful with what you choose to include and how you make relation to natural alternatives. I think it could be framed in a much different context to discuss examples like the Veterinary Feed Directive in the US or limits on antibiotics as growth promoters are being phased out if not already illegal.

Au: Thank you for your suggestion. We have re-edited the entire paragraph and accepted your context suggestion (L80-94).

Line 84: Change “over” to “on”.

Au: Thank you. We reformulate this paragraph to meet your suggestions.

Line 85-87: Mention of organic dairy farms is extraneous and should be removed. This statement has no bearing on overall objective of this paper.

Au: Thank you for your suggestion. We agree that this statement is not the main objective of this paper, so we remove it.

Line 91: change “on” to “with”

Au: Edited, thank you (L98).

Line 93: change “for” to “to”

Au: Edited, thank you (L100).

Line 94: ‘Leucocyte” should be “leukocyte”

Au: Edited, thank you (L101).

Line 95: Suggest “Lastly, essential oils have been shown to function similarly to ionophores by influencing…”

Au: Edited, thank you (L101-102).

Line 100: change “over” to “on”

Au: Edited, thank you (L107).

Line 106: the word ‘influence’ alone is unclear; was positive or negative influence hypothesized?

Au: We apologize for the misunderstanding. We reformulated the sentence to make clear the statement. (L118)

Materials and Methods

Line 120-121: ‘up to’ should be ‘before’ 

Au: Edited, thank you (L134)

Line 122: “At 2 to 3 d of age” should be “From 2 to 3 d of age” Correct? You fed transition milk on d 2 and 3?

Au: Edited, thank you. We fed colostrum on the first day, transitional milk on the second and third, and started with the milk replacer on the fourth. (L136)

Line 125, 200: the g should be italicized also is this g force value correct? I think it is rpm. It is different than other locations throughout materials and methods. Please verify.

Au: Thank you for your concern. Edited. The relative centrifugal force (RFC) is generated when a particle is subjected to a circular motion. The unit of measurement of the RCF is g, which is equivalent to the acceleration of gravity on the Earth's surface. Thus, the speed of a centrifuge will be provided in RCF (or g-force) or rotations per minute (rpm). (L140, 230). 

Line 127: you state that only heifers with a high enough Brix value were enrolled. 

Au: Thank you for your concern. Yes, only heifers with high Brix were enrolled (L142-143)

Were there any heifers that were not enrolled because of this criteria?

Au: Thank you for your concern. Only one heifer did not meet the Brix criteria, so she was taken out of the trial. Since we were evaluating animal’s performance as well, immune function and health, it was vital that all animals had had a proper colostrum management. Therefore, the results obtained would not be influenced by that variable.

Line 129-130: was intake recorded of water and starter? When were these first offered?

Au: Thank you for your concern. Water and starter intake values, both from pre-weaning and post-weaning, are available in table 2. All animals receive ad libitum water and starter during pre-weaning. And ad libitum water and 3 kg of starter on post-weaning. Water and starter were available since day 0 of the trial (L144-146).

Line 132: recommend “On d 4 of age, heifers were assigned to one of two experimental milk replacers. Both milk replacers were fed at a rate of 5 L/d…” May be able to reorganize and consolidate text you have on line 135. Feeding rate could be described after description of treatment milk replacers.

Au: Thank you for your contribution. We reformulated the paragraph (L147-163). However, we did not say that there were two different milk replaces since they were the same. The difference was that in the BEO treatment, we added the blend of essential oil additive.

Line 135: replace “with” with “at”

Au: Thank you for your concern. We edited the paragraph (L147-163).

Line 136- you state serum total protein but you actually measured Brix correct?

Au: Thank you for your contribution. Sorry for the misunderstanding. We corrected the sentence and other places of the text. It was measured Brix value to evaluate the success of the passive immune transfer (L148).

Line 137: How was dosage of 1g/calf/d determined? Is it manufacturer recommendation? Suggest addition of “or a commercial blend of essential oils supplemented at a rate of 1 g/calf/d (BEO; Apex.)

Au: Edited, thank you (L153). We used the manufacturer's recommendation. We also add that information to the text (L154).

Line 138: Delete “Apex Calf” from the second half of that line. Recommend “The BEO is a dry powder…”

Au: Edited, thank you (L154). We choose to keep “blend of essential oils” not to confuse with the experimental group abbreviation.

Line 140: essential oil “for” each meal

Au: Edited, thank you (L156).

Line 141: milk or milk replacer? Were calves fed the control diet fed and treated in a similar way or did they receive their whole meal at one time? Did this influence timing of feeding between treatments that might influence blood measurements collected relative to feeding?

Au: We apologize for the misunderstanding. We reformulate the sentence to make clear the statement (L157). All animals received milk replacer. Feeding time was at the same time. Animals were feed from the youngest to the oldest, always obeying this order. We also colored code the feeding buckets and washed them separately to avoid product residue to be feed to an animal, not in the treatment group. Feeding all animals would not last more than half an hour. So, we do think that did not influence blood measurements since we also did sampling from the youngest to the oldest. Animals from BEO treatment received 0.5L mixed with the blend of essential oil, and a bucket was put in front of their stall with the other 2.0 L. One person was responsible for refilling the milk bucket after they finished. Animals from the control group received the 2.5 L all at once. However, since we would refill the BEO group as soon as they finish, there were no differences in time and amount of milk replacer intake.

Table 1: Is this analyzed composition? Table heading should stand alone, suggest addition of “fed to calves”. You said you took samples for nutrient analysis but did not present standard deviations of feed ingredients. “unless otherwise noted” can be deleted. Extra space between % of in Organic matter heading. Was starch measured? It would be a good addition to analysis. Superscript 1 is not required. Superscript 2- if commercial milk can you provide brand name and company? Starter ingredients I think it is useful to know but not sure if entire list is necessary in this case.

Au: Thank you for your concern. We did not present the standard deviations od the ingredients since we have noticed in other Plos One papers’ this is not a normal practice (Pone.0234610 and Pone.0179940).

Unfortunately, we did not measure starch. Due to COVID-19, the laboratories and research facilities are closed, not allowing us to and it. 

We have taken superscript 1 out. The commercial milk replacer name is provided in the text, but we added on subscription. We decided to leave the entire list so readers can check starter quality.

Line 153 and 158: starter is listed to have monensin and a probiotic additive. How does this impact your treatments? Any consideration for this when interpreting your results? In intro you discuss ionophores… but do not say anything further.

Au: Thank you for your concern. We wanted to give a good quality starter and a starter used by farmers. Monensin is an important and efficient growth promoter antibiotic used to prevent coccidiosis and increase performance. Therefore, most of the starters used in Brazil today contain this additive. Since both treatments received the starter containing monesin, and there were no differences for starter intake, we think that this did not impact our results. It shows that there was no antagonism between both additives. It must be highlight also that since monsein was provided at the start, it would go to the rumen. The essential oils blend was provided in the milk replacer so that it would go to the intestines. Thus, the target for the action of both additives would be different. We have cited ionophores in the intro since it is expected that the essential oils present similar results. However, we think more studies must evaluate the comparison between essential oils and other additives, as well as the evaluation of interactions of the combined use of both molecules (L170-177).

Line 180: inconsistent use of “body development” and “structural growth”. I think structural growth more appropriate for the measures you collected.

Au: Thank you for your contribution. We revised the paper and change body development to structural growth (L198).

Line 181: “birth date” should be “day of birth”. BW measurements every 3 d needs a bit more clarification. How was this handled for analysis? Did the day BW was collected continually change day of week?

Au: Thank you for your concern. We chose to measure the BW every 3 days so that way we would have a more accurate measurement of the weight gain and over diseases impact over it. The measurement was calculated using the day of the calf birth and adding 3 more days after every weight. For that reason, there was not a specific day to weight the calves. That would vary according to its day of birth (L 199).

Line 192: Introduction states that rumen fluid was collected every 2 wk; however, collection days listed do not follow a 14-day pattern. Days listed are 14, 28, 42, 60, 74, and 90; a 14-day pattern would be 14, 28, 42, 56, 70 and 84. If samples were collected in a different timeframe than originally stated, please discuss. It seems as though the later collection dates were adjusted to account for collection on days that calves were weaned (60d) and also at the end of the collection period (90).

Au: Thank you for your concern. We adjusted to the main text to fit a better explanation. The sample collection was adjusted on the days that the calves were weaned. So, the pattern was 14, 28, 42, 60, 74, and 90 (L210).

Line 198: spelling error; ‘termo’ should be “Thermo”. Add “WI” after “Madison.”

Au: Edited, thank you (L216).

Line 206- insert “3 h after morning feeding” after collected

Au: Edited, thank you (L225).

Line 207: “feed” should be “ingestion”

Au: Edited, thank you (L224).

Blood samples not collected in weekly increments as stated in abstract; collection days after baseline collection at day 4 follow 7 day pattern until d 63, then are collected 11, 7 and 9 d later instead of consistently 7. Correct throughout to represent what was actually done.

Au: Thank you for your contribution. We revised the paper and saw that we specify the wrong days. We have taken a blood sample on day 0 that worked as a baseline before treatment. The sample of day 4 was used only to check blood Brix value. For the following days, we collected every 7 days and on the weaning day. For the post-weaning period, animals were collected based on the weaning day, which was day 60. Since the last blood collection on day 88, but animals would be out of the trial on day 90, we have pushed that blood sampling for 2 days. We corrected (L225-227).

Line 209: Collection days again do not match up to the initial 2-week timeframe stated for IGF-1 determination. Fluctuates between 7 and 14 day collection. Correct this throughout

Au: Thank you for your contribution. Also, as cited above, there was a miscommunication with the exact days of blood sampling. Samples for IGF-1 were taken on the day of birth as a baseline, and after that, every 14 days. Since day 56 was close to day 60 (the weaning day), it was chosen just to use the sample form day 60 since it would change that much. Sampling on the post-weaning phase took the weaning day as a basis, and for the same reason that happened during the pre-weaning phase, we choose to use the sampling that would be on day 88 and push it to day 90 (L226-227).

Line 222: essay should be assay

Au: Edited, thank you (L239).

Line 227: perform should be evaluate or analyze

Au: Edited, thank you (L244).

Line 235: delete “it was also performed” and add “were calculated” at the end of the sentence.

Au: Edited, thank you (L252).

Line 244: here you indicate classification of severe diarrhea but terminology used in results and discussion is different so could you clarify between the two sections of text what “severity” of diarrhea you are discussing please?

Au: Thank you for your concern. We classify diarrhea as fecal scores 2 and 3. However, when the fecal score 3 is watery, the animal losses more electrolytes and water, and it was considered to be severe. We added the explanation from the material and methods to the results and discussion (L262).

Line 255: “Final respiratory score considered the sum of all punctuations” should be reworded to “A final respiratory score was determined by the summation of all scores”. However could you clarify if diarrhea score was included in this or not? Maybe list which scores when into the final sum?

Au: Thank you for your contribution. We clarified that the respiratory score is calculated by the sum of temperature, cough, nose, eye, and ear scores, and the fecal score does not enter the final sum (L273-274).

Line 257: What antibiotic was used, what dosage, how long?

Au: Thank you for your contribution. We added the type and dosage of the antibiotic. Since it was a long-lasting drug, it was used only one dosage (L275-276),

Line 258 and 261: “subsequently” should be sequential

Au: Edited, thank you (L277).

Line 260: What is ‘pulmonary commitment’?

Au: Thank you for your concern. When an animal developed respiratory signs such as nose and eye discharge, cough, and fever, we would do pulmonary auscultation to check if we could find some shortness of breath, edema, and crepitation. Does were sighs of pulmonary commitment. We added that explanation to the text (L280).

Line 263: Consider moving this section to earlier in materials and methods section and discuss more about why it was used. Was this used to determine dosage?

Au: Thank you for your contribution. We perform the minimum inhibitory concentration to understand how the essential oils were acting on the changes of the fecal and respiratory scores, as well as the blood cells. However, this assay and results of it did not influence the given dosage. The idea of doing it came after the trial with the calves was already finished (L283).

Line 267: No footnote number after author reference; inconsistent with remainder of manuscript and instructions to authors

Au: Edited, thank you (287). 

Line 269: is the tween from a specific company?

Au: Added, thank you. It was from Sigma-Aldrich in Brazil (L289-290).

Line 290: Weren’t all the animals of similar genetic composition? Why was this used as a blocking effect?

Au: Thank you for your contribution. Gyr x Holstein is a typical cross dairy breed in Brazil. However, although the animals can have a similar genetic composition is still a very new breed with some variation among animals. For that reason, to have a more precise evaluation, we decided to block the genetic composition (L310).

Line 294-295: Rephrase “accept or deny”

Au: Edited, thank you (315). 

Line 297-298: were mean values that were transformed presented as back transformed values? SE would generally not be presented with transformed values. It would be a confidence interval 

Au: Thank you for or concern. We decided to add the CI at the bottom of the table for the transformed value (L352).

Results and discussion

Line 307: Remove ‘s’ from ‘first’.

Au: Edited, thank you (L327).

First paragraph or in R and D: what about timing of major enteric or other diseases that calves experience and timing of occurrence? Thinking about diarrhea in calves occurring mainly in first 3 wk of life and limited starter intake during this time which would support inclusion in milk if to benefit this situation. What about timing and main use of antibiotics relative to these health events? That is a main motivation for use of essential oils correct?

Au: Thank you for your concern. We added the timing of diarrhea occurrence. Farms usually use antibiotics to treat diarrhea, even if they do not know the cause. The primary motivation for the use of essential oils is to decrease disease morbidity and the use of antibiotics. Both disease, and antibiotics, can cause disorders on the gut microbiota and, consequently, to the animal itself. (L 327-328, and 332)

Line 311-312: how was this determined? Not described

Au: Thank you for your concern. We evaluated if the animals would refuse the milk replacer supplemented with the essential oils. We added this explanation to the material and methods (L156-163).

Line 316: Total mixed ration instead of total mixed ratio

Au: Edited, thank you (L341).

Line 319: consider changing ‘supply’ to ‘delivery method’ to be more clear

Au: Edited, thank you (L337).

Line 332-333: there is no description of a correlation in stats section. Please include in Stats section

Au: Included, thank you (L320-322).

Line 333: spelling error; ‘reveled’ should be ‘revealed’

Au: Edited, thank you (L359).

334-335: impacted on smaller should be resulted in reduced

Au: Edited, thank you (L361).

Line 344: EO was not defined previously and this sentence was already stated in the second paragraph of this section. Reorganize and consolidate

Au: Edited, thank you (L370).

349-350: this sentence does not make sense

Au: We apologize for the misunderstanding. We reformulate the sentence to make clear the statement (L375-377). 

Line 352: Table 3 should be placed here to remain consistent with rest of document and journal requirements

Au. Edited. Thank you (L392)

Line 356: Italicize P

Au. Edited. Thank you (L402)

Table 3: your SEM seem like it would result in a difference between treatments for BW at weaning and final BW are you underpowered to be able to detect these differences?

Au: Thank you for your concern. The test was not underpowered. We run a power test to verify if it was underpowered using pwr package and we had the following results:

For weaning body weight:

pwr.t.test(d = (66.66- 64.66)/1.07, power = .9, sig.level = .05, type = "two.sample", alternative ="two.sided")

Two-sample t test power calculation

n = 7.128182

d = 1.869159

sig.level = 0.05

power = 0.9

alternative = two.sided

NOTE: n is number in *each* group

For final body weight:

pwr.t.test(d=(89.88-93.34)/1.57, power = .9, sig.level = .05, type = "two.sample", alternative = "two.sided")

Two-sample t test power calculation

n = 5.491875

d = 2.203822

sig.level = 0.05

power = 0.9

alternative = two.sided

NOTE: n is number in *each* group

We also calculated the confidence interval:

Item Treatment SEM P – value

 CON

(n = 15) BEO

(n = 14) T

 Birth BW (kg) 32.40 (28.26 - 35.68) 31.97 (28.88 - 35.91) 0.59 0.85

 Weaning BW (kg) 63.82 (57.51 - 70.14) 67.03 (60.35 -73.70) 1.07 0.45

 Final BW (kg) 89.88 (80.60 - 99.14) 93.34 (83.54 - 103.13) 1.57 0.57

ADG should be kg/d and feed efficiency should be kg/kg

Au: Thank you. Edited 

Any initial measure of body weight or structure used for covariate for respective variable?

Au: Thank you for your concern. The birth body weight was used as a covariate to analyze weight and biometrics.

Line 373: ‘present’ should be ‘presented’

Au. Edited. Thank you (L415)

Line 376: how was “heifers’ ingestion behavior” evaluated?

Au: Thank you for your concern. We apologize for the misunderstanding. We reformulate the sentence to make clear the statement. What we suggest is that the heifers from BEO group could have a higher intake of starter in the morning when it was available, and that could lower the ruminal pH. Since we would evaluate intake just after 24 hours, we did not know how that intake was distributed throughout the day. So, it was evaluated the amount of intake and not intake behavior. For intake behavior, we mean when and how much at a time the heifer would eat (L418 – 423).

Line 379: spaces between numbered citations

Au. Edited. Thank you (L422, 423).

Line 388: reference for this statement?

Au. Added. Thank you (L433).

Line 395: “Besides the previous cited effects” is vague. Define a little more specifically in terms of health benefits, etc.

Au: Thank you for your concern. We apologize for the misunderstanding. We reformulate the sentence to make clear the statement (L410-411).

Line 397: Simply saying that essential oils can cause a toxic effect could insinuate a negative connotation to animal health; reword to relate better to its detriment on harmful bacteria rather than to the rumen, which is how it reads as currently written.

Au: Thank you for your concern. We apologize for the misunderstanding. We reformulate the sentence to make clear the statement (L439, 440).

Line 399: Similar thoughts on use of the word ‘consequence’.

Au: Thank you for your concern. We apologize for the misunderstanding. We reformulate the sentence to make clear the statement (L444).

Line 409-410: rewrite this section it is really confusing. Group without EO would be CON correct?

Au: Thank you for your concern. We apologize for the misunderstanding. We reformulate the sentence to make clear the statement (L455, 456).

Line 410-415: what about positive benefits to small intestine on whole GIT development? There is a little bit of information on this or theories at least.

Au: Thank you for your contribution. We added a statement of Meale et al. 2017 about the connection of the lower gut and forestomach (L464-467).

Line 415: Not sure of use of extravasation; considering its definition it doesn’t fit how you’re using it

Au: Thank you for your concern. We apologize for the misunderstanding. We reformulate the sentence to make clear the statement (L464 - 473).

Line 419-428: This paragraph does not contribute a lot to the overall results; very weak correlations between references and findings. Context is needed from other references cited to add more meaning to the discussion. Not sure it fits here

Au: Thank you for your concern. We apologize for the misunderstanding. We reformulate the sentence to make clear the statement (L474 - 481).

Line 458: remove ‘of’ before ‘insulin’. What are the references for these conclusions?

Au. Added. Thank you (L511)

Line 460: remove ‘s’ from ‘these’.

Au. Edited. Thank you (L513)

Table 6. define PLR and NLR

Au: Thank you for your concern. The definition is at the bottom of the table (L536-537).

Line 503-506: were these measured in your study? Connection to your results?

Au: Thank you for your contribution. We did not measure does variables, and that statement did not add anything to the discussion. For that reason, we decided to take all the sentence out.

Line 511: and/or

Au. Edited. Thank you (L565)

Line 523: what cell count?

Au: Thank you for your concern. We apologize for the misunderstanding. We reformulated the sentence. What we did mean is that treatments were made from the days that blood was collected to run a hemogram (L578 -579).

Line 525: health score is respiratory score?

Au: Thank you for your contribution. It is the same thing, but it was edited so that it would be referred to as a respiratory score only (L 584).

Table 7: when was temperature or health evaluated daily? Could you add this to materials and methods

Au: Thank you for your concern. We started the health measurements explanation on material and methods saying that scores were performed daily, in the morning, before any kind of animal management (L256- 257).

I don’t think full score explanation is need again. Reference McGuirk

Au. Edited. Thank you.

Line 554-556: Absolutely cannot draw a correlation between or make an assumption about respiratory signs and diarrhea.

Au: Thank you for your concern. We apologize for the misunderstanding. We reformulated the sentence. We did not evaluate the direct correlation of the enteric cases and respiratory cases. So, you are right that we cannot make this assumption. However, since both diseases are multifactorial and have an association already cited in the literature, we suppose that there is a correlation. (L599-603).

Line 571: spelling error: ‘trough’ should be ‘through’

Au. Edited. Thank you (L618)

Line 566-579: Should include potential effects due to monensin and probiotic additives in starter on results here. If you did not see an in vitro effect at the concentration of 1.0 µg/mL, would that not suggest that you consider a different dosage?

Au: Thank you for your concern. Since both treatments received the same starter, the effects due to the monensin and the probiotic additives would be for all animals in this trial, and not just for one treatment. So, having those additives do not concern us about the trial results.

We did the in vitro study to see if we could find a direct effect of the essential oils’ molecules over bacteria. Apparently, this product does not show direct action over the tested bacteria. However, the supplementation with essential oils on animals change some immunological values, and those could explain differences found in the fecal score. Using different dosages and routes of supplementation could help us clarify some results in this paper. 

Line 586- rout should be route

Au. Edited. Thank you (L634)

Figure 1 has labels that are cutoff, need units for variables in stead of “value”, “Trat” should be treatment

Au: Thank you. Edited.

Figure 2 lymphocytes is spelled incorrectly

Au: Thank you. Edited.

Reviewer #2:

ABSTRACT

Consider reporting the specific P-value instead of a general P ≤ 0.001 or P ≤ 0.05.

Au: Edited, thank you (L 53-55).

MATERIALS AND METHODS

Line 140: essential oils instead of essential oil.

Au: Edited, thank you (L154). 

Line 293: define performance. Does it mean BW and body measures?

Au: Thank you for your contribution. As suggested by the first reviewer, we change to structural growth (L 313).

RESULTS AND DISCUSSION

Similar to the abstract section, consider reporting the specific P-value in the text instead of a general P ≤ 0.001 or P ≤ 0.05.

Au: Thank you for your contribution. We made all changes throughout the text.

Line 334: inverse association…, were. Did you mean where?

Au: Thank you for your contribution. We apologize for the misunderstanding. We reformulated the sentence. What we wanted to mean was that when the fecal scores were high, the milk replacer intake was low. Thus, they have a negative correlation (L 359 - 361). 

Line 341 – 351: consider adding of discussing why the lack of difference in responses in your study compared with those cited.

Thank you for your contribution. We changed the discussion and added some possibilities of why there was a lack of difference (L378-390).

The reviewer considers that the authors should present the fatty acid composition or profile, at least the major fatty acids present in the BEO. It is a fundamental analysis to include in the manuscript and accounts for in the results and discussion section.

Au: Thank you for your concern about it. According to Benchaar et al. (2008), the name “essential oil” comes from “essence,” related to the property that these substances of providing flavor and odors, and “oil” since they are mostly arranged with low-density lipid composts. However, they are not true oils since they are not made of fatty acids. The essential oils are blends of plant’s secondary chemical complex, characterized as a volatile aromatic compound, with 20 to 60 different chemical substances.

---

## [Decision Letter · Decision Letter 1]

21 Sep 2020

PONE-D-20-07435R1

Effects of a blend of essential oils in milk replacer on performance, rumen fermentation, blood parameters, and health scores of dairy heifers

PLOS ONE

Dear Dr. Palhares Campolina,

Thank you for submitting your manuscript to PLOS ONE. After careful consideration, we feel that it has merit but does not fully meet PLOS ONE’s publication criteria as it currently stands. Therefore, we invite you to submit a revised version of the manuscript that addresses the points raised during the review process.

THERE ARE SOME MINOR ISSUES THAT STILL NEED TO BE FIXED.

We look forward to receiving your revised manuscript.

Kind regards,

Juan J Loor

Academic Editor

PLOS ONE

Reviewers' comments:

Reviewer's Responses to Questions

**Comments to the Author**

1. If the authors have adequately addressed your comments raised in a previous round of review and you feel that this manuscript is now acceptable for publication, you may indicate that here to bypass the “Comments to the Author” section, enter your conflict of interest statement in the “Confidential to Editor” section, and submit your "Accept" recommendation.

Reviewer #1: (No Response)

Reviewer #2: All comments have been addressed

2. Is the manuscript technically sound, and do the data support the conclusions?

Reviewer #1: Yes

Reviewer #2: Yes

3. Has the statistical analysis been performed appropriately and rigorously? 

Reviewer #1: Yes

Reviewer #2: Yes

4. Have the authors made all data underlying the findings in their manuscript fully available?

Reviewer #1: No

Reviewer #2: Yes

5. Is the manuscript presented in an intelligible fashion and written in standard English?

Reviewer #1: Yes

Reviewer #2: Yes

6. Review Comments to the Author

Reviewer #1: Overall, the majority of the edits were completed by the authors and have made improvements to the manuscript. There are still a few areas that need to be modified or clarified to be resolved and are detailed below. Some conclusions and supporting citations do not necessarily fit with some of your results.

Line 53: delete “in”

Line 56-57: replace “to” with “in” after both “ruminal manipulation” and “carryover effects”.

Line 61-63: Suggest “A good calf rearing program should embrace aspects that encompass body development, stress reduction, meet nutritional requirements, and housing management to optimize calf health status.”

Line 64: suggest “essential” should be “key”

Line 70: “Raising” should be “raise”

Line 73: Please include reference after statement regarding rumen development beginning with “Additionally”.

Line 75: Grammar edit- “too” should be “to”; also add “to” after “adapt”.

Line 80: “Wildly” should be “widely.”

Lines 80-88: Antibiotic growth promoters and antibiotics used for treatment of acute onset of illness are not the same. The statement that 90% of dairy farms provide AGPs for disease prevention (Line 86) is not correct, for several reasons. First, the publication from which you reference this percentage uses data collected from the mid to late 2000s, and was published prior to the USDA’s Veterinary Feed Directive in 2015. Under the VFD, use of AGPs is no longer permitted on US dairy farms without Veterinary supervision (nor in most other countries). Second, according to the literature referenced, this number refers to the percentage of dairy farms in the US that administer antibiotics at dry-off to prevent intramammary infection during the dry period- which is not the same as an AGP. I also caution you about your reference to 80% of all antibiotics used in the country are from livestock production (Line 84); this number was calculated by ‘public health advocacy groups’ and is cautioned against in the publication by the FDA. Many glaring inconsistencies exist in data attempting to find the source of antibiotic overuse, and the reality is that judicious use of antimicrobials needs to happen across all health sectors, not just one. Also not recommended to use the word “abuse” (Line 88); suggest ‘overuse’ instead.

Suggest revising and truncating this paragraph to focus on the motivation to find alternatives to antibiotic use due to growing concerns of resistance and ineffectiveness, and structure it more to lead into the exploration of phytogenic agents as alternatives.

Line 92: extra space between “first line”

Line 95:- Deleted “The” at the beginning of the sentence

Line 100: “change” should be “by changing”

Line 101-102: reference for this statement?

Line 103-104: “Improving” should be “improve”; “decreasing” should be “decrease”.

Line 105: Change “focus” to “focusing”. Add apostrophe after “oils”. Consider doing additional search to find more data on essential oil supplementation in swine and poultry to further supplement this section if you want to continue to make comparison.

Line 108: Remove apostrophe from “molecules” and place on “oils”.

Line 113: Add “if” after “evaluate”.

Line 134: “hours” should be “h”

Line 140: “minutes should be “min”

Line 141: Change “pipped” to “piped”.

Line 151: delete “New Zealand” already stated above so don’t need to state again.

Line 166 Table 1 and line 189: If you have weekly composite samples of each feedstuffs for analysis you should include or have a standard deviation for your nutrient analysis. Please include or restate how you analyzed your feedstuffs for this experiment. Add to Statistical Analysis section

Line 181: should be “fixed for a maximum…”

Line 184: add “treatment” after anti-inflammatory.

Line 188: “was” should be “were”

Line 189: “week” should be “weekly”

Line 199: Start sentence with “Body weight”

Line 206: Sentence is incomplete.

Line 224 to 225: delete “after that,”

Line 252: suggest “calculated [27]. In addition, platelet… (NLR) were calculated”

Line 262: Remove “severe” as it is redundant.

Line 277: Suggest adding “consecutive” after two and delete sequential

Line 283-294: It is unclear why a 1 microgram/mL concentration was chosen as your concentration to test the MIC. If my calculations are correct 1 g of BEO= 1,000,000 micrograms. When 1,000,000 micrograms are divided by the fluid volume of milk replacer provided per day to each calf (5,000 mL) this is 200 micrograms per ML for the dosage rate, yet you tested 1 microgram/mL. Could you explain in more detail why the concentration that was used for MIC was done and how this relates to the your actual feeding rate in the experiment. Suggest discussion in your results and discussion section.

Line 293: extra space after 35

Line 315: Italicize P in P-value

Line 330-331: Suggest “compromised” should be “limited and “pre-weaning” might be better suited to describe as first 30 d of life when intake of starter will be low so the desired supplementation level may not be achieved based on intake levels of the starter.

Line 331: suggest deleting “it was decided to offer” and add “was offered” after BEO

Line 333: “consequently” should be “subsequently”

Line 334: suggest deleting sentence starting with “there was no rejection” and replacing “with no refusal and good acceptance” with “indicating no palatability issues of BEO”

Line 337: delete “way of”

Line 340: What do you mean by “young animals x old animals”?

Line 344: I think you need to end this paragraph by restating that in this study palatability was not a problem with the mixture used.

Line 348 and 394- Don’t need n= 29 because of number below in table.

Table 2- for MR intake in because of how you analyzed and presented the CI I don’t think SEM should be included in the table.

Line 358- Suggest “An observed effect between fecal scores…” Although I am not sure this correlation really adds to your point. It seems like your objective is to determine a treatment by time interaction of your variables in response to BEO which can be described for both MR intake and fecal scores already without a further correlation between the two. As a reader I think I would be ok with you stating that MR intake was lower during certain weeks (if significant) and also fecal scores were elevated during those same weeks which are likely related because intake decreases when animals are sick.

Line 367- according to table 3 feed efficiency in the preweaning period tended to be lower in calves supplemented with BEO but no difference postweaning. On Line 375 you make about postweaning feed efficiency carry over effects but there does not appear to be any carryover effects in terms of intake or feed efficiency in your experiment. It is unclear what connections and conclusions you are trying to make here.

Line 380: Correct spelling of “monensin”

Line 383: Replace “masquerade” with “mask”

Line 384-385: Suggest “In this study, no antagonism between additives was observed, as there were no negative responses for BEO compared to CON”

Table 3: “Kg/Kg” should be “kg/kg” for both feed efficiencies

Line 403-406: Your comment about structural growth being related to protein in the diet in starter- how does this relate to protein provided (MR and starter) and consumed by your calves related to their expected requirements or growth. By bringing in this information are you saying your calves had already met their requirements for optimal growth or that they were limited on protein so you might not expect a response?

Line 410: “developed” instead of “develop”

Line 417: Wrong P-value listed in text; this is the pre-weaning week effect P-value. Your actual P-value for this variable was only a tendency so make sure you adjust your wording and conclusions related to this appropriately.

422-424- it is unclear what this sentence contributes

Table 4- The C2:C3 ratios presented in the table seem high based on the Acetic and Propionic values presented for example preweaning C2 was 30.8 and C3 was 18.88 which is a ratio of 1.63 not 1.97. Could you verify the numbers you presented are correct?

Line 436-439- I am not sure what these points really contribute to understanding what happened in response to the treatments you used and the results you collected

Line 44--: how are essential oils enhanced by low pH?

Line 446- your results don’t support decreased ruminal nitrogen ammonia or increases in propionate and butyrate

Line 456 to 457- it is not clear the values that you are presenting in this statement or how they relate to what you did. Are these values form a different experiment? What does the P-value relate to? If it is a different study you shouldn’t report the actual P-value. Need more context.

Line 461- “This way of providing it” suggest should be “By providing the BEO in the MR, the treatment should be diverted past the rumen and have minimal impact on local ruminal microbiota and VFA.”

Line 465- delete “to”

Line 468: “it is shown a direct effect” should be “supplementation of essential oils have shown a direct effect”

Line 483- Suggest” For all ruminal parameters, a week effect during preweaning was observed”

Line 515- Suggest “All blood cell counts were within normal ranges based on age and species

Line 521- delete “are” and “originated” should be originate

Line 526 to 527 and line 529 to 530: need context about which treatment you are talking about

Line 546: delete “An”

Line 562: Reference needed

Line 567: BEO shouldn’t be used here because that is your treatment description whereas this is a generalization of other blends of essential oils. Reference for this statement?

Line 572: add P-value for days with diarrhea. And Percent of calves with diarrhea (Line 572-574)

Line 575: should be changed over time

Line 576: “within” should be “between” and add P-value

Line 581: Replace “it is hard to point out” with “it is important to point out” and remove “since”.

Table 7: Pre-weaning period is listed as 4-60 d previously; change 61 to 60 for consistency.

Line 596: what about avg days for CON?

Line 616: Capitalize Gram.

Reviewer #2: The reviewer thanks the authors for addressing the comments raised in the previous review. However, there are few more comments or suggestions to improve the presentation of the manuscript.

7. PLOS authors have the option to publish the peer review history of their article (what does this mean?). If published, this will include your full peer review and any attached files.

Reviewer #1: No

Reviewer #2: No

---

## [Author Response · Author response to Decision Letter 1]

14 Oct 2020

Responses to reviewers

Reviewer #1: Overall, the majority of the edits were completed by the authors and have made improvements to the manuscript. There are still a few areas that need to be modified or clarified to be resolved and are detailed below. Some conclusions and supporting citations do not necessarily fit with some of your results.

Line 53: delete “in”

Au: Edited. Thank you (L. 53)

Line 56-57: replace “to” with “in” after both “ruminal manipulation” and “carryover effects”.

Au: Edited. Thank you (L. 56-57).

Line 61-63: Suggest “A good calf rearing program should embrace aspects that encompass body development, stress reduction, meet nutritional requirements, and housing management to optimize calf health status.”

Au: Thank you for your suggestion. Edited (L.61-63).

Line 64: suggest “essential” should be “key”

Au: Edited. Thank you (L. 64).

Line 70: “Raising” should be “raise”

Au: Edited. Thank you (L. 70).

Line 73: Please include reference after statement regarding rumen development beginning with “Additionally”.

Au: Thank you for your concern. We included the reference in the statement (L.75).

Line 75: Grammar edit- “too” should be “to”; also add “to” after “adapt”.

Au: Edited. Thank you (L. 75).

Line 80: “Wildly” should be “widely.”

Au: Edited. Thank you (L. 80).

Lines 80-88: Antibiotic growth promoters and antibiotics used for treatment of acute onset of illness are not the same. The statement that 90% of dairy farms provide AGPs for disease prevention (Line 86) is not correct, for several reasons. First, the publication from which you reference this percentage uses data collected from the mid to late 2000s, and was published prior to the USDA’s Veterinary Feed Directive in 2015. Under the VFD, use of AGPs is no longer permitted on US dairy farms without Veterinary supervision (nor in most other countries). Second, according to the literature referenced, this number refers to the percentage of dairy farms in the US that administer antibiotics at dry-off to prevent intramammary infection during the dry period- which is not the same as an AGP. I also caution you about your reference to 80% of all antibiotics used in the country are from livestock production (Line 84); this number was calculated by ‘public health advocacy groups’ and is cautioned against in the publication by the FDA. Many glaring inconsistencies exist in data attempting to find the source of antibiotic overuse, and the reality is that judicious use of antimicrobials needs to happen across all health sectors, not just one. Also not recommended to use the word “abuse” (Line 88); suggest ‘overuse’ instead.

Suggest revising and truncating this paragraph to focus on the motivation to find alternatives to antibiotic use due to growing concerns of resistance and ineffectiveness, and structure it more to lead into the exploration of phytogenic agents as alternatives.

Au: Thank you for your concern. We did an intense literature review and we re-wrote the entire paragraph. Your comment is extremely pertinent, and we changed the previous statements. Thank you for your contribution (L.80-103).

Line 92: extra space between “first line”

Au: Thank you for your concern. With the paragraph review, this sentence has been changed. 

Line 95:- Deleted “The” at the beginning of the sentence

Au: Edited. Thank you (L. 104).

Line 100: “change” should be “by changing”

Au: Edited. Thank you (L. 110).

Line 101-102: reference for this statement?

Au: Thank you for your concern. We added the reference for the statement (L. 111).

Line 103-104: “Improving” should be “improve”; “decreasing” should be “decrease”.

Au: Edited. Thank you (L. 113).

Line 105: Change “focus” to “focusing”. Add apostrophe after “oils”. Consider doing additional search to find more data on essential oil supplementation in swine and poultry to further supplement this section if you want to continue to make comparison.

Au: Edited. Thank you. We also added additional data of essential oils use for swine and poultry (L. 114-121).

Line 108: Remove apostrophe from “molecules” and place on “oils”.

Au: Edited. Thank you (L. 123).

Line 113: Add “if” after “evaluate”.

Au: Edited. Thank you (L. 127).

Line 134: “hours” should be “h”s

Au: Edited. Thank you (L. 148).

Line 140: “minutes should be “min”

Au: Edited. Thank you (L. 153).

Line 141: Change “pipped” to “piped”.

Au: Edited. Thank you (L. 154).

Line 151: delete “New Zealand” already stated above so don’t need to state again.

Au: Edited. Thank you (L. 164).

Line 166 Table 1 and line 189: If you have weekly composite samples of each feedstuffs for analysis you should include or have a standard deviation for your nutrient analysis. Please include or restate how you analyzed your feedstuffs for this experiment. Add to Statistical Analysis section

Au: Thank you for your concern. We calculated the weekly composite samples' standard deviations and added to the table (L 177).

Line 181: should be “fixed for a maximum…”

Au: Edited. Thank you (L. 193).

Line 184: add “treatment” after anti-inflammatory.

Au: Edited. Thank you (L. 196).

Line 188: “was” should be “were”

Au: Edited. Thank you (L. 200).

Line 189: “week” should be “weekly”

Au: Edited. Thank you (L. 201).

Line 199: Start sentence with “Body weight”

Au: Edited. Thank you (L. 212).

Line 206: Sentence is incomplete.

Au: Thank you for your concern. We added the missing part of the sentence (L. 219).

Line 224 to 225: delete “after that,”

Au: Edited. Thank you (L. 237).

Line 252: suggest “calculated [27]. In addition, platelet… (NLR) were calculated”

Au: Edited. Thank you (L. 266- 267).

Line 262: Remove “severe” as it is redundant.

Au: Edited. Thank you (L. 275-276).

Line 277: Suggest adding “consecutive” after two and delete sequential

Au: Edited. Thank you (L. 291).

Line 283-294: It is unclear why a 1 microgram/mL concentration was chosen as your concentration to test the MIC. If my calculations are correct 1 g of BEO= 1,000,000 micrograms. When 1,000,000 micrograms are divided by the fluid volume of milk replacer provided per day to each calf (5,000 mL) this is 200 micrograms per ML for the dosage rate, yet you tested 1 microgram/mL. Could you explain in more detail why the concentration that was used for MIC was done and how this relates to the your actual feeding rate in the experiment. Suggest discussion in your results and discussion section.

Au: Thank you for your concern. Your calculations are right, and the dose used in the MIC was wrongly expressed. We wanted to say that we started with an initial concentration of 1.0 mg/mL (Five times higher than the one used for the calves), and then we diluted it from 1:2 until 1:256. We changed to the correct measure unit. We are sorry for the misunderstanding (L. 305).

Line 293: extra space after 35

Au: Edited. Thank you (L. 307).

Line 315: Italicize P in P-value

Au: Edited. Thank you (L. 329).

Line 330-331: Suggest “compromised” should be “limited and “pre-weaning” might be better suited to describe as first 30 d of life when intake of starter will be low so the desired supplementation level may not be achieved based on intake levels of the starter.

Au: Thank you for your concern. The suggestion makes the statement clearer. Edited (L. 344-346).

Line 331: suggest deleting “it was decided to offer” and add “was offered” after BEO

Au: Edited. Thank you (L. 346).

Line 333: “consequently” should be “subsequently”

Au: Edited. Thank you (L. 348).

Line 334: suggest deleting sentence starting with “there was no rejection” and replacing 

“with no refusal and good acceptance” with “indicating no palatability issues of BEO”

Au: Edited. Thank you (L. 349).

Line 337: delete “way of”

Au: Edited. Thank you (L. 351).

Line 340: What do you mean by “young animals x old animals”?

Au: Thank you for your concern. We changed the statement to make a clear statement. By young animals, we mean juvenile animals, and old animals, we mean adults (L. 307).

Line 344: I think you need to end this paragraph by restating that in this study palatability was not a problem with the mixture used.

Au: Thank you for your concern. We added a sentence at the end of the paragraph to emphasize the response as you suggested (L. 359-360).

Line 348 and 394- Don’t need n= 29 because of number below in table.

Au: Edited. Thank you (L. 364 and 409).

Table 2- for MR intake in because of how you analyzed and presented the CI I don’t think SEM should be included in the table.

Au: Edited. Thank you. We removed the SEM (L. 365).

Line 358- Suggest “An observed effect between fecal scores…” Although I am not sure this correlation really adds to your point. It seems like your objective is to determine a treatment by time interaction of your variables in response to BEO which can be described for both MR intake and fecal scores already without a further correlation between the two. As a reader I think I would be ok with you stating that MR intake was lower during certain weeks (if significant) and also fecal scores were elevated during those same weeks which are likely related because intake decreases when animals are sick.

Au: Thank you for your concern. Besides, it is well known that intakes decrease with sickness, we did the correlation just to statistically prove that the association between intake and the fecal score was negative (L. 373 and 375).

Line 367- according to table 3 feed efficiency in the preweaning period tended to be lower in calves supplemented with BEO but no difference postweaning. On Line 375 you make about postweaning feed efficiency carry over effects but there does not appear to be any carryover effects in terms of intake or feed efficiency in your experiment. It is unclear what connections and conclusions you are trying to make here.

Au: Thank you for your concern. The statement on line 375 referend to carry-over effects in other studies and not on this one. However, we reformulate the sentence to be clear. (L. 391-394).

Line 380: Correct spelling of “monensin”

Au: Edited. Thank you (L. 397).

Line 383: Replace “masquerade” with “mask”

Au: Edited. Thank you (L. 400).

Line 384-385: Suggest “In this study, no antagonism between additives was observed, as there were no negative responses for BEO compared to CON”

Au: Edited. Thank you (L. 401-403).

Table 3: “Kg/Kg” should be “kg/kg” for both feed efficiencies

Au: Edited. Thank you.

Line 403-406: Your comment about structural growth being related to protein in the diet in starter- how does this relate to protein provided (MR and starter) and consumed by your calves related to their expected requirements or growth. By bringing in this information are you saying your calves had already met their requirements for optimal growth or that they were limited on protein so you might not expect a response?

Au: Thank you for your concern. We formulated the diet and increased the solids in the milk replacer to meet the requirement to our calves for optimal growth. However, we did not test different protein levels in the starter or in the milk replacer to see their interaction with the essential oil supplementation. We added a statement to the manuscript to clarify this point (L.424-426). 

Line 410: “developed” instead of “develop” 

Au: Edited. Thank you (L. 429).

Line 417: Wrong P-value listed in text; this is the pre-weaning week effect P-value. Your actual P-value for this variable was only a tendency so make sure you adjust your wording and conclusions related to this appropriately.

Au: Thank you for your concern. We corrected the p-value on the text. We considered statistically different when P ≤ 0.05, and we did not work with tendencies (L. 437).

422-424- it is unclear what this sentence contributes

Au: Thank you for your concern. The sentence was relocated to a different place to contribute with the result discussion (L. 429).

Table 4- The C2:C3 ratios presented in the table seem high based on the Acetic and Propionic values presented for example preweaning C2 was 30.8 and C3 was 18.88 which is a ratio of 1.63 not 1.97. Could you verify the numbers you presented are correct?

Au: Thank you for your concern. We re-run the statistics and the values are correct. The results not presented in the manuscript are listed below.

C2:C3

Treatment emmean SE df lower.CL upper.CL

 BEO 1.69 0.147 23 1.38 1.99

 CON 1.97 0.133 23 1.69 2.24

Week

 3 2.59 0.227 23 2.13 3.06

 5 1.88 0.169 23 1.53 2.23

 7 1.48 0.146 23 1.18 1.78

 9 1.35 0.151 23 1.04 1.67

SEM = 0.119

CV = 17

Acetic

Treatment emmean SE df lower.CL upper.CL

 BEO 27.16 2.012 23 22.99 31.32

 CON 30.79 2.050 23 26.55 35.04

Week

 3 17.15 2.838 23 11.17 22.92

 5 30.35 2.358 23 25.47 35.23

 7 33.27 2.022 23 29.09 37.45

 9 35.25 2.100 23 30.91 39.60

SEM = 8.15

CV = 25.9

Propionic

Treatment emmean SE df lower.CL upper.CL

 BEO 20.01 1.455 23 17.00 21.65

 CON 18.18 1.339 23 16.11 23.02 

Week

 3 8.50 2.460 23 3.41 13.59

 5 19.20 2.275 23 14.49 23.90

 7 23.76 1.178 23 21.32 26.20

 9 26.33 1.291 23 23.66 29.00

SEM = 7.11

CV = 33.2

Line 436-439- I am not sure what these points really contribute to understanding what happened in response to the treatments you used and the results you collected

Au: Thank you for your concern. We out the statement just to justify why our animals had low pH when compared to adult animals and that this low ruminal pH did not create any health problem.

Line 44--: how are essential oils enhanced by low pH?

Au. Thank you for your question. Some studies reported the there is a higher level of bioactivity at acidic pH. So, at low pH the essential oils behave in a more hydrophobic way and enter more easily inside the cells. This explains how the antimicrobial activity of the essential oils is enhanced at low pH.

Line 446- your results don’t support decreased ruminal nitrogen ammonia or increases in propionate and butyrate

Au: Thank you for your concern. Indeed, there were no statistical differences in ruminal ammonia or the individual VFA values. However, there was a statistical difference within the proportion between the VFA. Therefore, we believe that besides no statistical difference in the VFA alone, it could be a biological difference that could lead to the difference in the proportions.

Line 456 to 457- it is not clear the values that you are presenting in this statement or how they relate to what you did. Are these values form a different experiment? What does the P-value relate to? If it is a different study you shouldn’t report the actual P-value. Need more context.

Au. Thank you for your concern. This statement refers to another experiment. We have reformulated the statement to the manuscript to clarify this point (L.473-475). 

Line 461- “This way of providing it” suggest should be “By providing the BEO in the MR, the treatment should be diverted past the rumen and have minimal impact on local ruminal microbiota and VFA.”

Au: Thank you for your suggestion. Added (L.479-481)

Line 465- delete “to”

Au: Edited. Thank you (L. 484).

Line 468: “it is shown a direct effect” should be “supplementation of essential oils have shown a direct effect”

Au: Thank you for your suggestion. Added (L.487).

Line 483- Suggest” For all ruminal parameters, a week effect during preweaning was observed”

Au: Thank you for your suggestion. Added (L.503).

Line 515- Suggest “All blood cell counts were within normal ranges based on age and species

Au: Thank you for your suggestion. Added (L.535)

Line 521- delete “are” and “originated” should be originate

Au: Edited. Thank you (L.541).

Line 526 to 527 and line 529 to 530: need context about which treatment you are talking about

Au: Thank you for your suggestion. We added more information to make a clearer statement (L.547-548).

Line 546: delete “An”

Au: Edited. Thank you (L.566).

Line 562: Reference needed

Au: Added. Thank you (L.584).

Line 567: BEO shouldn’t be used here because that is your treatment description whereas this is a generalization of other blends of essential oils. Reference for this statement?

Au: Thank you for your concern. BEO is our treatment description. We changed the statement to be clearer. We added the reference. (L.587, 589)

Line 572: add P-value for days with diarrhea. And Percent of calves with diarrhea (Line 572-574)

Au: Added. Thank you (L.592 and 594).

Line 575: should be changed over time

Au: Edited. Thank you (L.595).

Line 576: “within” should be “between” and add P-value

Au: Edited. Thank you (L.596-597).

Line 581: Replace “it is hard to point out” with “it is important to point out” and remove “since”.

Au: Thank you for your suggestion. Edited. (L.602-603).

Table 7: Pre-weaning period is listed as 4-60 d previously; change 61 to 60 for consistency.

Au: Edited. Thank you (Table 7).

Line 596: what about avg days for CON?

Au: Thank you for your concern. The average was the same for both treatments. However, we added to make a clearer statement (L.617).

Line 616: Capitalize Gram.

Au: Edited. Thank you (L. 637).

Reviewer #2: The reviewer thanks the authors for addressing the comments raised in the previous review. However, there are few more comments or suggestions to improve the presentation of the manuscript.

 Manuscript Number PONE-D-20-07435

Effects of a blend of essential oils in milk replacer on performance, rumen

fermentation, blood parameters, and health scores of dairy heifers

The authors significantly improved the manuscript. After reviewing the new version of the manuscript, the reviewer has the following minor suggestions: 

ABSTRACT

Line 56: do not start sentences with abbreviations. The sentence could begin with The BEO…

Au: Edited. Thank you (L. 156).

MATERIALS AND METHODS

Line 56: do not start sentences with abbreviations, “The concentrations of NDF and ADF were determined…” instead. Similar on lines 194, 218, 232, 238, 493.

Au: Edited. Thank you (L. 206, 230, 244-245, 251, 506-507).

Line 157: 15 mL instead 15 ml.

Au: Edited. Thank you (L. 170).

Line 206: incomplete sentence…?

Au: Thank you for your concern. It was missing part of the sentence and we added to complete the statement (L. 218).

Line 227: remove the “d” after 90. 

Au: Edited. Thank you (L. 239).

Line 289: Tween 80 should be capitalized since it is the name of a commercial product. 

Au: Edited. Thank you (L. 303).

RESULTS AND DISCUSSION

Line 338 – 340: the sentence is difficult to follow. Consider paraphrasing it. 

Au: Thank you for your concern. We changed the sentence to make a clear statement (L. 352-354).

Line 341: it seems the word oil is missing after “cinnamaldehyde essential”

Au: Added. Thank you (L. 303).

Line 423: the words “treatment effect” at the end of the sentence seem to be redundant.

Au: Thank you for your concern. We have relocated this sentence and took the redundant words (L. 435-436).

---

## [Decision Letter · Decision Letter 2]

4 Dec 2020

PONE-D-20-07435R2

Effects of a blend of essential oils in milk replacer on performance, rumen fermentation, blood parameters, and health scores of dairy heifers

PLOS ONE

Dear Dr. Palhares Campolina,

Thank you for submitting your manuscript to PLOS ONE. After careful consideration, we feel that it has merit but does not fully meet PLOS ONE’s publication criteria as it currently stands. Therefore, we invite you to submit a revised version of the manuscript that addresses the points raised during the review process.

PLEASE ADDRESS CAREFULLY THE VARIOUS COMMENTS OF REVIEWER #1 IN A REVISED VERSION. THERE ARE STILL A NUMBER OF ISSUES THAT NEED FIXING.

We look forward to receiving your revised manuscript.

Kind regards,

Juan J Loor

Academic Editor

PLOS ONE

Reviewers' comments:

Reviewer's Responses to Questions

**Comments to the Author**

1. If the authors have adequately addressed your comments raised in a previous round of review and you feel that this manuscript is now acceptable for publication, you may indicate that here to bypass the “Comments to the Author” section, enter your conflict of interest statement in the “Confidential to Editor” section, and submit your "Accept" recommendation.

Reviewer #1: (No Response)

Reviewer #2: All comments have been addressed

2. Is the manuscript technically sound, and do the data support the conclusions?

Reviewer #1: Partly

Reviewer #2: Yes

3. Has the statistical analysis been performed appropriately and rigorously? 

Reviewer #1: No

Reviewer #2: Yes

4. Have the authors made all data underlying the findings in their manuscript fully available?

Reviewer #1: Yes

Reviewer #2: Yes

5. Is the manuscript presented in an intelligible fashion and written in standard English?

Reviewer #1: Yes

Reviewer #2: Yes

6. Review Comments to the Author

Reviewer #1: The authors have made effort to improve their manuscript. This is an interesting topic that warrants investigation. However, there are still several areas throughout where revisions are required. I think care should be taken when comparing different essential oils as the mechanism of action can be quite different.

No description of weaning strategy.

Figures are not included in latest submission so would not be able to see if they corrected spelling in Figure 2 of the axis for “Linphocytes”

No explanation was explained for the incomplete data from a calf (8424) in the raw data.

Randomized block design was stated but in the experimental model. Only genetic composition of the animal was ultimately used as a blocking effect while it was stated that genetics, birth month, birth BW, and % Brix were all used to balance treatments.

When data has been transformed the P-values should be presented should relate to the transformed analysis, the means should be back transformed, and a confidence interval should be presented instead of the SEM to aid in interpretation of the data presented. This was not provided for milk replacer intake.

I have an issue with the discussion of palatability in the discussion section. I do not believe that this was a true test of the palatability of the additive because it was dosed in a small portion of the milk replacer diet and calves were not given the rest of their milk replacer allotment until the initial dose had been consumed. While palatability is an important question to ask I think the authors make too bold of conclusions based on this experiment and do not address the limitations of how this was conducted and evaluated.

Discussion about acidosis and low pH is not related to the objectives and do not add to the understanding of how the BEO additive impacted the animal in this experiment (line 451-457)

The discussion on impact on the rumen is contradictory because at times the authors make the point that it the treatment additive was fed in the milk replacer and should bypass the rumen but go on to make major conclusions based on observed significance related to rumen fermentation changes.

Line 81- “promoters (AGP) have been”

Line 85- “The overuse of antimicrobial’s concern”

Line 87- Insert “use of” before “AGP”

Line 91- delete “;” and insert “and”

Line 95- insert “the acceptability” before AGP’s

Line 100- replace “revel” with “appear”

Line 110- “changing” should be “change”

Line 118- delete “on”

Line 177- Add “± SD” after DM basis

Line 267- “was” should be “were”

Line 291- “two consecutive days”

Line 304- “mL”

Line 344-345- delete “especially during the first month of life when the starter intake is low,”

Question related to palatability- The feeding of BEO in this study was done in a small portion of the milk replacer each day and the full feeding was not given until after that had been consumed? When trying to interpret effects of BEO addition on palatability I would be cautious or maybe incorporate this point into the discussion.

Line 359- Suggest “cinnamon as part of the mixture in our study”

Table 2- if Confidence intervals were used those should be reported in the table.

Line 442- insert “once” before every

Line 472 to 474- this sentence is confusing because I think you are saying the essential oil supplemented calves would have a lower C2:C3 ratio (1.56 and 1.47) compared to the control groups (2.02 and 1.77) but the order of how it is currently written is the opposite.

Line 616- suggestion deletion of “1.0 d for” and only say BEO and CON.

Line 619- insert a space after “=”

Reviewer #2: (No Response)

7. PLOS authors have the option to publish the peer review history of their article (what does this mean?). If published, this will include your full peer review and any attached files.

Reviewer #1: No

Reviewer #2: No

---

## [Author Response · Author response to Decision Letter 2]

14 Jan 2021

Reviewer #1: The authors have made effort to improve their manuscript. This is an interesting topic that warrants investigation. However, there are still several areas throughout where revisions are required. I think care should be taken when comparing different essential oils as the mechanism of action can be quite different.

No description of weaning strategy.

Au: Thank you for your concern. The weaning was performed abruptly because it is the management used on the farm. We added this statement to the text (L.191).

Figures are not included in latest submission so would not be able to see if they corrected spelling in Figure 2 of the axis for “Linphocytes”

Au: Thank you. We already corrected the figures and the spelling. We checked in the submission system and the figures were there. We will check with the editors if there is an error to understand why they were not available to you. We submit them again in this new version. 

No explanation was explained for the incomplete data from a calf (8424) in the raw data.

Au: Thank you for your concern. We are terribly sorry for the misunderstand. During the experiment, the data was saved in different excel spread sheets. To turn it available we united all in just one sheet. During that process, 8424 information after day 74 was not added and we did not see it. We revised all data and add the missing information.

Randomized block design was stated but in the experimental model. Only genetic composition of the animal was ultimately used as a blocking effect while it was stated that genetics, birth month, birth BW, and % Brix were all used to balance treatments.

Au: Thank you for your concern. We acknowledge that there was a mistake in the way that this statement was written, thus we rewrote to make it clearer. We used only genetic composition as a blocking effect. The animals were randomly assigned to the treatments. Birth month, birth BW and % Brix were assessed only to check if the groups were balanced. (L. 30- 32; 322-325).

When data has been transformed the P-values should be presented should relate to the transformed analysis, the means should be back transformed, and a confidence interval should be presented instead of the SEM to aid in interpretation of the data presented. This was not provided for milk replacer intake.

Au: Thank you for your concern. The confidence interval for that outcome was added at the bottom of the table. However, since it was not noticeably clear and easy to visualize, we added to the table (L. 177).

I have an issue with the discussion of palatability in the discussion section. I do not believe that this was a true test of the palatability of the additive because it was dosed in a small portion of the milk replacer diet and calves were not given the rest of their milk replacer allotment until the initial dose had been consumed. While palatability is an important question to ask I think the authors make too bold of conclusions based on this experiment and do not address the limitations of how this was conducted and evaluated. 

Au: Thank you for your concern. This is a good perspective and we also think is important to point the limitations of our study. We choose the wrong word to express what we wanted to say. Therefore, we added the study limitations and change the word palatability for ingestibility (L.359-364).

Discussion about acidosis and low pH is not related to the objectives and do not add to the understanding of how the BEO additive impacted the animal in this experiment (line 451-457)

Au: Thank you for your concern. We agree that this statement does not add any information and is not related with the objectives of the paper. Therefore, we decided to exclude the entire paragraph.

The discussion on impact on the rumen is contradictory because at times the authors make the point that it the treatment additive was fed in the milk replacer and should bypass the rumen but go on to make major conclusions based on observed significance related to rumen fermentation changes.

Au: Thank you for your concern. We make different points about what could have happened. The essential oils were fed in the milk replacer. If they were fed in the starter, changes in the rumen would be obvious and easy to explain. However, since they were fed in the milk replacer, they should bypasss the rumen to the abomasum and intestines. Therefore, one theory would be that the esophageal groove was still one and the oils could have entered the rumen. Other theory it about the potential connections between the intestinal tract. Those are all hypothesis. And to make a clearer statement we have changed the paragraph (L. 473 - 484).

Line 81- “promoters (AGP) have been”

Au: Edited. Thank you (L.81).

Line 85- “The overuse of antimicrobial’s concern”

Au: Edited. Thank you (L.85).

Line 87- Insert “use of” before “AGP”

Au: Edited. Thank you (L.86-87).

Line 91- delete “;” and insert “and”

Au: Edited. Thank you (L.91).

Line 95- insert “the acceptability” before AGP’s

Au: Added. Thank you (L.95).

Line 100- replace “revel” with “appear”

Au: Edited. Thank you (L.100).

Line 110- “changing” should be “change”

Au: Edited. Thank you (L.109).

Line 118- delete “on”

Au: Deleted. Thank you (L.118).

Line 177- Add “± SD” after DM basis

Au: Added. Thank you (L.177).

Line 267- “was” should be “were”

Au: Edited. Thank you (L.267).

Line 291- “two consecutive days”

Au: Edited. Thank you (L.291).

Line 304- “mL”

Au: Edited. Thank you (L.304).

Line 344-345- delete “especially during the first month of life when the starter intake is low,”

Au: Deleted. Thank you.

Question related to palatability- The feeding of BEO in this study was done in a small portion of the milk replacer each day and the full feeding was not given until after that had been consumed? When trying to interpret effects of BEO addition on palatability I would be cautious or maybe incorporate this point into the discussion.

Au: Thank you for your concern. The rest of the meal was only given after the calf consumed all the MR containing the additive. Although we did not see any visual difference in the time of MR intake, we agree that we could not say that the calves did not have any palatability issue with the additive since this outcome was not tested. We changed our statement in the discussion section (L. 359 -364).

Line 359- Suggest “cinnamon as part of the mixture in our study”

Au: Thank you for your suggestion. We changed the statements in this section to make a clearer point of view and that sentence was deleted.

Table 2- if Confidence intervals were used those should be reported in the table.

Au: We added the confidence intervals to the table. Thank you for your suggestion (L.366) 

Line 442- insert “once” before every

Au: Added. Thank you (L.443).

Line 472 to 474- this sentence is confusing because I think you are saying the essential oil supplemented calves would have a lower C2:C3 ratio (1.56 and 1.47) compared to the control groups (2.02 and 1.77) but the order of how it is currently written is the opposite.

Au: Edited. Thank you. We changed the statement in the previous version and the values were put in the opposite place (L.468).

Line 616- suggestion deletion of “1.0 d for” and only say BEO and CON.

Au: Edited. Thank you (L.610).

Line 619- insert a space after “=”

Au: Added. Thank you (L.613).

Reviewer #2: (No Response)

---

## [Decision Letter · Decision Letter 3]

19 Feb 2021

Effects of a blend of essential oils in milk replacer on performance, rumen fermentation, blood parameters, and health scores of dairy heifers

PONE-D-20-07435R3

Dear Dr. Palhares Campolina,

We’re pleased to inform you that your manuscript has been judged scientifically suitable for publication and will be formally accepted for publication once it meets all outstanding technical requirements.

Kind regards,

Juan J Loor

Academic Editor

PLOS ONE

Additional Editor Comments (optional):

Reviewers' comments:

Reviewer's Responses to Questions

**Comments to the Author**

1. If the authors have adequately addressed your comments raised in a previous round of review and you feel that this manuscript is now acceptable for publication, you may indicate that here to bypass the “Comments to the Author” section, enter your conflict of interest statement in the “Confidential to Editor” section, and submit your "Accept" recommendation.

Reviewer #1: All comments have been addressed

2. Is the manuscript technically sound, and do the data support the conclusions?

Reviewer #1: Yes

3. Has the statistical analysis been performed appropriately and rigorously? 

Reviewer #1: Yes

4. Have the authors made all data underlying the findings in their manuscript fully available?

Reviewer #1: Yes

5. Is the manuscript presented in an intelligible fashion and written in standard English?

Reviewer #1: Yes

6. Review Comments to the Author

Reviewer #1: (No Response)

7. PLOS authors have the option to publish the peer review history of their article (what does this mean?). If published, this will include your full peer review and any attached files.

Reviewer #1: No

---

## [Editor Report · Acceptance letter]

25 Feb 2021

PONE-D-20-07435R3 

Effects of a blend of essential oils in milk replacer on performance, rumen fermentation, blood parameters, and health scores of dairy heifers 

Dear Dr. Palhares Campolina:

I'm pleased to inform you that your manuscript has been deemed suitable for publication in PLOS ONE. Congratulations! Your manuscript is now with our production department. 

Kind regards, 

on behalf of

Dr. Juan J Loor 

Academic Editor

PLOS ONE